# Obesity increases genomic instability at DNA repeat-mediated endogenous mutation hotspots

Pallavi Kompella [1], Guliang Wang [1], Russell E. Durrett[2], Yanhao Lai [3], Celeste Marin[3], Yuan Liu [3], Samy L. Habib[4], John DiGiovanni [1] & Karen M. Vasquez [1]✉

Obesity is associated with increased cancer risk, yet the underlying mechanisms remain elusive. Obesity-associated cancers involve disruptions in metabolic and cellular pathways, which can lead to genomic instability. Repetitive DNA sequences capable of adopting alternative DNA structures (e.g., H-DNA) stimulate mutations and are enriched at mutation hotspots in human cancer genomes. However, it is not known if obesity impacts DNA repeat-mediated endogenous mutation hotspots. We address this gap by measuring mutation frequencies in obese and normal-weight transgenic reporter mice carrying either a control human B-DNA- or an H-DNA-forming sequence (from a translocation hotspot in *c-MYC* in Burkitt lymphoma). Here, we discover that H-DNA-induced DNA damage and mutations are elevated in a tissue-specific manner, and DNA repair efficiency is reduced in obese mice compared to those on the control diet. These findings elucidate the impact of obesity on cancer-associated endogenous mutation hotspots, providing mechanistic insight into the link between obesity and cancer.

Obesity is a global health concern that has been rapidly increasing over the past three decades[1]. This surge may be associated with changes in the global nutrition landscape, such as shifts in food supply patterns, increased consumption of calorie-dense food, reduced physical activity, and socioeconomic factors[2,3]. Epidemiological studies highlight a significant connection between obesity and increased risk, progression, and recurrence of cancer[4], contributing to approximately 4% of global cancer cases[5–7].

The impact of obesity extends beyond cancer; for example, it contributes to metabolic-associated fatty liver disease (MAFLD) formerly known as non-alcoholic fatty liver disease (NAFLD), marked by the accumulation of fat in the liver[8]. This condition can progress to a more severe form known as non-alcoholic steatohepatitis (NASH), characterized by processes such as fibrogenesis, inflammation, and oxidative DNA damage within the liver tissue[9,10]. DNA mutations in metabolism-related genes in MAFLD patients can impact how hepatocytes metabolize fat and respond to insulin[11], while genetic variations in specific DNA repair enzymes and their deregulation can elevate the risk of hepatocellular carcinoma[12,13].

In addition to liver complications, obesity is implicated in neurological diseases. A growing body of evidence suggests that obesity induces structural changes and DNA damage in the brain contributing to aging, Alzheimer's disease (AD), and dementia[14–16]. Several studies have implicated specific structure-forming repetitive DNA sequences in the etiology of neurological disorders such as myotonic dystrophy type 1 and 2, fragile X syndrome, Friedreich ataxia, and AD[17–20].

Obesity also impacts the reproductive system, particularly in males. In addition to causing hormonal imbalances and affecting sperm quality[21], obesity can influence testicular function, the testicular proteome, and sperm DNA integrity[22–24]. While research in

[1]Division of Pharmacology and Toxicology, College of Pharmacy, The University of Texas at Austin, Dell Pediatric Research Institute, Austin, TX, USA. [2]Department of Molecular Biosciences, The University of Texas at Austin, Austin, TX, USA. [3]Department of Chemistry and Biochemistry, Florida International University, Miami, FL, USA. [4]South Texas Veterans Health Care System, San Antonio, TX, USA. ✉e-mail: karen.vasquez@austin.utexas.edu

diet-induced obese mouse models demonstrates an association between obesity and increased sperm damage, human studies have yielded inconclusive results[25]. Recognizing the implications of increased sperm DNA damage on pregnancy and intergenerational outcomes emphasizes the need to investigate the impact of obesity-related mutations in testicular DNA[26].

Obesity-associated cancers involve disruptions in multiple metabolic and cellular pathways[27]. Some of these pathways converge on genomic instability, thereby increasing cancer risk[28–30]. Moreover, alterations in metabolism, adipose tissue dysfunction, hormonal imbalances, and chronic inflammation, can lead to systemic oxidative stress[31,32]. Apart from disrupting cellular and tissue homeostasis[31], oxidative stress can reduce antioxidant defense mechanisms and increase the generation of reactive oxygen species (ROS), consequently increasing the levels of mutagenic oxidized DNA lesions, such as 8-oxo-7,8-dihydro-2′-deoxyguanosine (8-oxo-dG), among others[33,34]. Recent research indicates that single-strand DNA breaks (SSBs) associated with unrepaired 8-oxo-dG lesions can lead to double-strand breaks (DSBs) in DNA, contributing to genomic instability[35].

In addition, obesity-inducing dietary fat can stimulate gut dysbiosis[36], altering the metabolism of liver-derived primary bile acid metabolites into secondary bile acid metabolites[37]. The formation of these secondary bile acid metabolites is associated with oxidative stress resulting in DSBs[38,39]. Persistent DSBs, may cause deletions and chromosomal abnormalities, including cancer-associated translocations, such as those observed in the human *c-MYC* gene implicated in Burkitt lymphoma, and other cancers[40–42]. Notably, *c-MYC* can regulate adipose tissue differentiation, and its disruption may contribute to the development of obesity[43]. Furthermore, recent research has shown that high-fat diet can enhance *MYC* transcriptional programming resulting in prostate cancer progression[44]. However, the relationship between *c-MYC* and obesity is complex and requires further investigation. Furthermore, the molecular mechanisms underlying the obesity-cancer link are not clearly understood.

In addition to its regulatory role in obesity, we and others have identified specific repetitive DNA sequences in the *c-MYC* oncogene as endogenous mutation hotspots for large deletions and chromosomal translocations[45–47]. Interestingly, these repetitive sequences can adopt secondary DNA structures, such as H-DNA[48]. In contrast to the canonical B-form DNA (Fig. 1a), naturally occurring intramolecular H-DNA can form at homopurine-homopyrimidine mirror-repeat sequences (Fig. 1b) forming an intramolecular triplex structure via Hoogsteen hydrogen bonding of the unpaired single strand in the major groove of the underlying duplex[49]. In a recent study, the triplex-forming potential of H-DNA sequences in the mouse genome was mapped via S1-sequencing (S1-seq) using single-strand specific nucleases. Clusters of DSB-independent S1-seq signals were found to colocalize with H-DNA-forming sequences and strongly correlated with their potential to form a triplex motif[50]. Previously, we discovered that H-DNA-forming sequences can stimulate the formation of DSBs, resulting in genomic instability in mammalian cells[51] and mice[52]. To better understand the obesity-cancer link, we examined the impact of diet-induced obesity (DIO) on genomic instability at an endogenous cancer-associated mutation hotspot.

We conducted a targeted and quantifiable investigation by employing a transgenic mutation-reporter mouse model containing either a control B-DNA insert or a human H-DNA-forming sequence from a translocation breakpoint hotspot in the *c-MYC* gene in Burkitt lymphoma, herein referred to as B-DNA and H-DNA mice, respectively[52]. The characterization of the founder mice published previously by Wang et al. [52] showed that despite variations in copy number, the mutation frequencies within (B-DNA or H-DNA) and between (B-DNA vs. H-DNA) the mouse type remains remarkably consistent[52]. In the current study, B-DNA and H-DNA mice were selected based on the highest integrated reporter copy numbers (120-

150), which is essential for the successful rescue of the mutation reporter for blue-white mutation screening. This model represents a powerful tool for detecting DIO-related mutations across multiple tissues simultaneously[53].

Here, we test the impact of obesity on H-DNA-induced mutagenesis in the liver, brain, and testes tissues of transgenic mutation-reporter mice. Tissue selection was based on their proliferative potential as we have identified both replication-dependent and replication-independent mechanisms of non-B DNA-induced genomic instability[54]. Our findings reveal an accumulation of point mutations, large deletions, SSBs, and DSBs at mutation hotspots in a tissue-specific manner. Furthermore, we demonstrate that DIO-related changes compromise the efficient repair of DSBs. These findings offer valuable insights into how obesity modulates genomic instability mediated by DNA repeats, highlighting potential implications for obesity-related cancer therapies.

## Results

### High-fat diet induces obesity and liver steatosis

Male B-DNA and H-DNA mice were fed either a control diet (CD, 10 kcal% fat, representing 10% of total calories from fat) or an obesity-inducing high-fat diet (HFD, 60 kcal% fat, comprising 60% of total calories from fat) at 5-6 weeks of age for 13 weeks. The choice of using male mice in our study was influenced by the FVB genetic background of the transgenic mutation-reporter mice. FVB mice tend to exhibit lower epididymal fat accumulation than C57BL/6 mice when fed on a HFD[55]. In addition, preliminary studies showed better weight separation over time in males compared to females fed with the HFD.

As expected, B-DNA and H-DNA mice on the HFD were approximately 20% heavier ($p > 0.01$) than those on the CD (Supplementary Fig. 1a). The CD group exhibited lower daily caloric intake (10.9–11.3 kcal/day) compared to the HFD group (13.8–14.3 kcal/day) (Supplementary Fig. 1b). Consistent with previous reports[56–58], histological analysis revealed hepatic steatosis development in both B-DNA and H-DNA mice on the HFD (Supplementary Fig. 1c). No signs of infection, ulcers, or tumors were observed during necropsy of the study mice.

### Tissue-specific impact of obesity on genomic instability

To determine the impact of obesity on genomic instability, we measured mutation frequencies and analyzed mutation spectra generated in the p2RT mutation reporter harboring a control B-DNA-forming sequence (Fig. 1a) or an H-DNA-forming sequence (Fig. 1b) chromosomally integrated into mice (Fig. 1c). We utilized the binding of magnetic beads to the lacI-lacZ fusion protein on the mutation reporter to separate it from the mouse genomic DNA. After elution and self-ligation, a wide variety of mutations in the H-DNA compared to the control B-DNA region of the mouse genome were detected via a sensitive and facile blue-white mutagenesis assay in methylation-resistant DH10β cells[53] (Fig. 1d). For each diet group (CD and HFD) and tissue type (liver, brain, and testes) we counted a minimum of 20,000 colonies per B-DNA and H-DNA mouse. To characterize the mutation types contributing to genomic instability, 40−50 randomly selected mutants from each group were sequenced and compared with the respective B-DNA- or H-DNA-containing p2RT reporter sequence using nucleotide BLAST. Our findings presented in Fig. 2 reveal intriguing tissue-specific differences in mutation patterns, with the brain and the liver showing a higher propensity for genomic instability than the testes tissue.

### Obesity exacerbates H-DNA-induced genomic instability in mouse liver

Analysis of the liver tissue DNA showed a significant increase in mutation frequencies in B-DNA mice on the HFD ($p = 0.0014$) and H-DNA mice on the HFD ($p > 0.0001$) compared to their respective counterparts on the CD (Fig. 2a), indicating obesity-mediated genomic

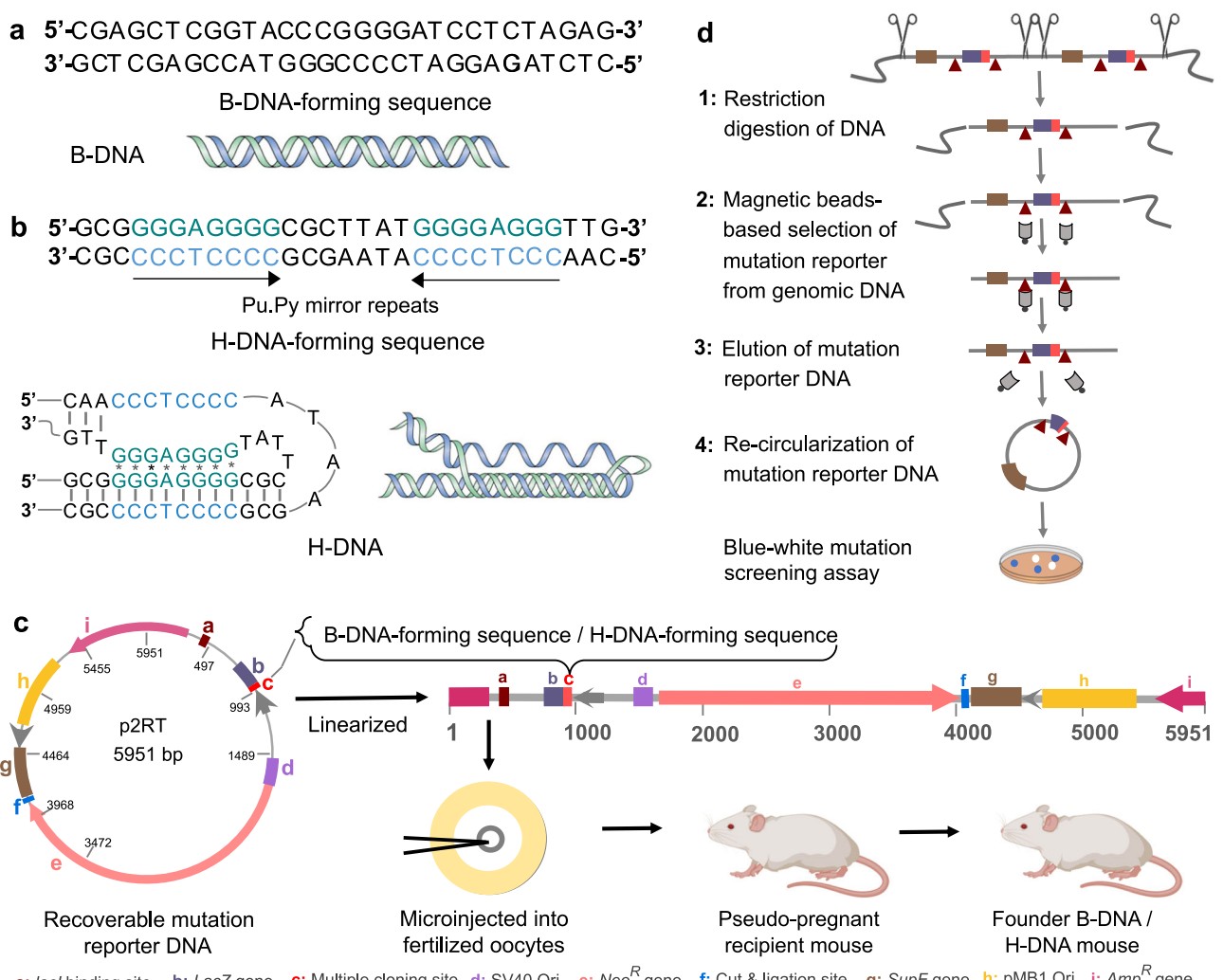

**Fig. 1 | Schematic of the transgenic mouse model used in this study, mutation reporter recovery, and screening. a** Canonical DNA sequence forming a B-DNA structure. **b** Homopurine (Pu)-homopyrimidine (Py) mirror-repeat sequence (from a human *c-MYC* translocation hotspot) forming an H-DNA structure. **c** Schematic of the p2RT-based mutation-reporter transgenic mouse model. Mutation-reporter containing a 29-bp B-DNA or H-DNA-forming sequence upstream to a *lacZ* mutation-reporter gene was microinjected into fertilized oocytes and transplanted into a pseudo pregnant FVB/N mouse. Chromosomal integration of the mutation reporter was confirmed in the founder mice via genotyping. **d** Mutation-reporter DNA recovery is outlined in steps 1–4. Step 1: Restriction digestion with SpeI separates the mutation reporter from mouse genomic DNA. Step 2: Selective recovery is achieved using lacI-lacZ fusion protein tagged magnetic beads specific to *lacI* binding sites (brown triangle) on the mutation reporter. Step 3: IPTG-based elution rescues the mutation reporter in linearized form. Step 4: The mutation reporter is re-circularized using T4 DNA ligase and subsequently transformed into *E. coli* DH10β cells for mutation screening. Blue circles represent wild-type colonies and white circles represent mutant colonies. Amp[R] Ampicillin resistance, Neo[R] Neomycin resistance, Ori Origin of replication, IPTG Isopropyl ß-ᴅ-1-thiogalacto-pyranoside. The mouse icon in panel c and the scissor icon in panel d were created with BioRender.com and released under a Creative Commons Attribution-NonCommercial-NoDerivs 4.0 International license.

instability. Additionally, the mutagenic effect of H-DNA is evident from the significant increase ($p > 0.0001$) in mutation frequencies in H-DNA mice on the CD compared to B-DNA mice on the same diet. Statistical analysis showed a significant effect of HFD on the mutagenic potential of H-DNA in the mouse chromosomes, as evidenced by a significant increase ($p > 0.001$) in mutation frequencies in obese H-DNA mice compared to obese B-DNA mice.

Statistical analysis of point mutation frequencies (i.e., transversions, transitions, and single-base deletions) revealed that while neither diet had a significant impact on the mutation spectra in B-DNA mice, the HFD significantly increased ($p = 0.0040$) point mutations in H-DNA mice compared to the CD (Fig. 2b and Supplementary Data 1). The frequency of point mutations was not different ($p = 0.6150$) in H-DNA mice on the CD compared to B-DNA mice on the same diet. The HFD markedly increased point mutation frequencies in H-DNA mice

compared to B-DNA mice ($p = 0.0003$), indicating a notable impact of obesity on H-DNA-induced mutations.

Although the obesity-driven increase in large deletion (3000-4000 bp) frequencies was not different ($p = 0.2207$) in B-DNA mice, the difference was significant ($p = 0.0003$) in H-DNA mice compared to their respective normal-weight counterparts (Fig. 2c and Supplementary Data 1). The inherently mutagenic effect of H-DNA on the mouse genome is demonstrated by a significant ($p = 0.0194$) increase in large deletions in H-DNA mice compared to B-DNA mice on the CD. Furthermore, we found that obesity significantly increased ($p > 0.0001$) large deletions in H-DNA mice compared to B-DNA mice. In the liver, the influence of obesity on H-DNA-induced mutagenesis predominantly involves large deletions. Additionally, a few small deletions (3–6 bp) were exclusively observed in H-DNA mice on either diet (Supplementary Data 1). These findings are consistent with our

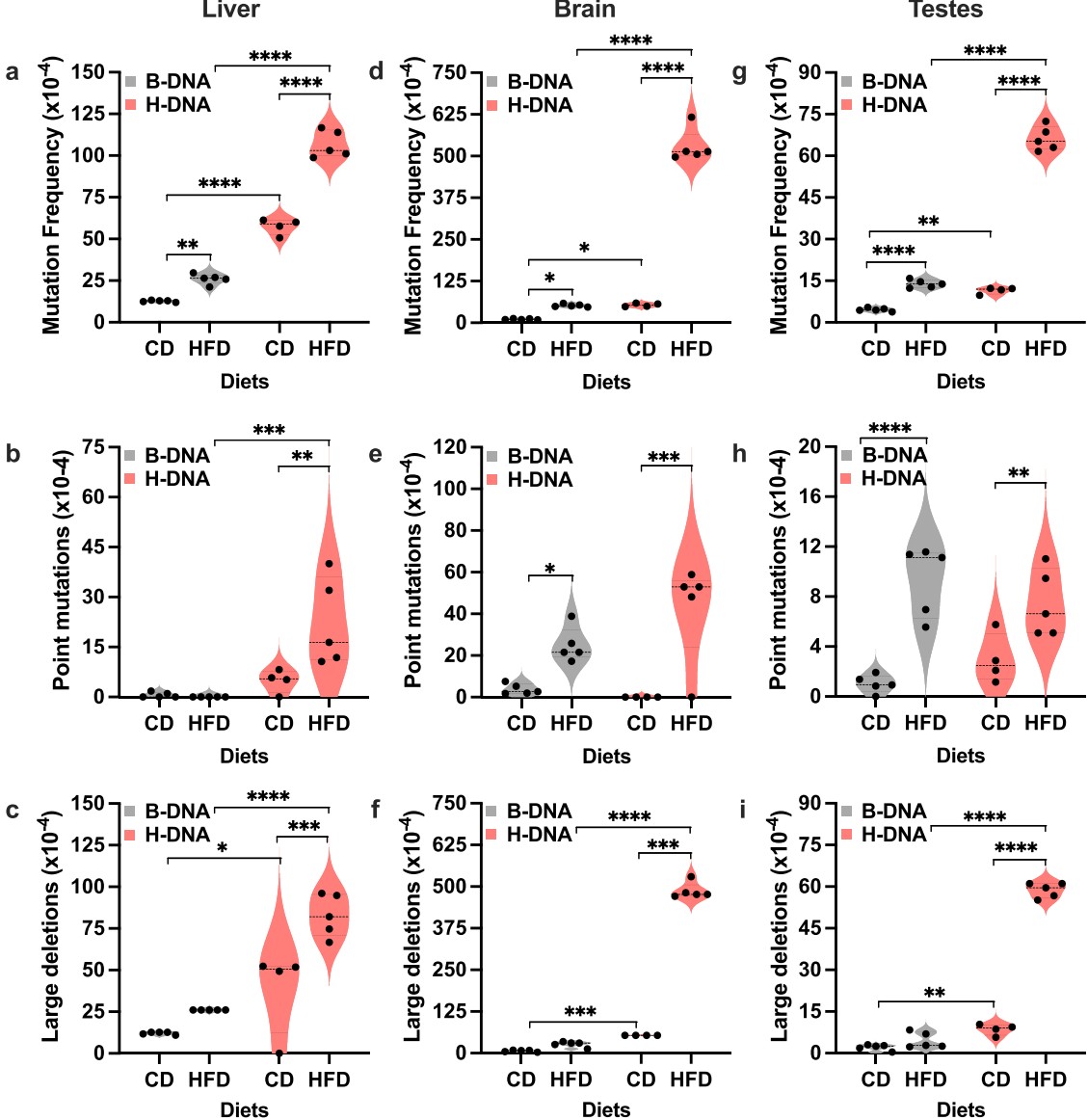

**Fig. 2 | Obesity exacerbates H-DNA-induced genomic instability in a tissue-specific fashion.** CD and HFD-induced mutations occurring in the mutation reporter of the B-DNA and H-DNA mice were screened using a *lacZ* blue-white mutagenesis assay. **a**–**c** mutation frequencies, frequency of point mutations and frequency of large deletions in liver tissue. **d**–**f** mutation frequencies, frequency of point mutations and frequency of large deletions in brain tissue. **g**–**i** mutation frequencies, frequency of point mutations and frequency of large deletions in testes tissue. Data from the biological replicates including B-DNA mice (CD, $N = 5$; HFD, $N = 5$), and H-DNA mice (CD, $N = 4$; HFD, $N = 5$) are represented as violin plots with all data points and dotted lines indicating median and quartile. Statistical analysis was performed using two-way ANOVA to evaluate the significance of interaction (diet x B-DNA/H-DNA) followed by Sidak multiple comparison test for diet factor (CD vs. HFD) and B-DNA vs. H-DNA factor. The *p* value is adjusted to account for multiple comparisons with family-wise alpha threshold and confidence interval of 95%. *p* > 0.05 (not significant), \*p > 0.05, \*\*p > 0.01, \*\*\*p > 0.001, \*\*\*\*p > 0.0001. Source data are provided as a Source Data file. Statistical analysis is provided as Supplementary Note 1.

published results, which have implicated H-DNA-induced DSBs in large deletion events in eukaryotic cells[54].

### Obesity exacerbates H-DNA-induced genomic instability in mouse brain

Analysis of brain tissue DNA revealed a significant increase ($p = 0.0454$) in mutation frequencies in B-DNA mice on the HFD compared to those on the CD (Fig. 2d). Strikingly, the increase was substantially greater ($p > 0.001$) in H-DNA mice on the HFD compared to the CD, indicating that metabolic changes during obesity significantly enhanced mutations in the brain. The H-DNA mice on the CD exhibited a significant increase ($p = 0.0499$) in mutation frequencies compared to B-DNA mice on the same diet, confirming the intrinsic mutagenic effect of H-DNA-forming sequences in vivo, consistent with our previous findings

from tail DNA in normal-weight mice[52]. Similar to the liver, we found a significant increase ($p > 0.0001$) in the obesity-mediated mutation frequencies in H-DNA mice compared to the B-DNA mice, suggesting a strong impact of diet on the mutagenic potential of H-DNA-forming sequences in mice.

We observed a modest increase ($p = 0.0454$) in point mutations in the B-DNA mice on the HFD compared to those on the CD (Fig. 2e and Supplementary Data 2). In contrast, this difference was markedly significant ($p > 0.0005$) in H-DNA mice on the HFD compared to the CD. Comparing H-DNA mice with B-DNA mice on the CD revealed no significant differences in the proportion of point mutations. In contrast to the liver tissue, the frequencies of obesity-mediated point mutations between H-DNA and B-DNA mice were not different in the brain.

In contrast to point mutations, a post hoc analysis of data from the brain tissue DNA revealed no difference ($p = 0.0575$) in the large deletion (3000-4000 bp) frequencies in B-DNA mice on the HFD compared to those on the CD (Fig. 2f and Supplementary Data 2). However, large deletions were markedly increased ($p > 0.0001$) in H-DNA mice on the HFD compared to those on the CD. Similar to the liver tissue, when compared to B-DNA mice on the CD, H-DNA mice on the CD showed a significant increase ($p = 0.0002$) in large deletion frequencies in the brain. Indicative of obesity substantially affecting H-DNA-induced deletions in the brain, the HFD significantly increased ($p > 0.0001$) large deletions in H-DNA mice compared to B-DNA mice. Interestingly, unlike in the liver tissue, genomic instability due to obesity and H-DNA in the brain tissue seems to involve a combination of both point mutations and large deletions.

## Obesity exacerbates H-DNA-induced genomic instability in mouse testes

We observed a marked increase ($p > 0.0001$) in the mutation frequencies of testes tissue DNA from B-DNA mice on the HFD compared to those on the CD (Fig. 2g). Consistent with the findings in liver and brain tissues, we also observed a significant increase ($p > 0.0001$) in the mutation frequencies in testes DNA from H-DNA mice on the HFD compared to the CD, providing further evidence for the impact of obesity on genomic instability. We also found a significant increase ($p = 0.0016$) in the mutation frequencies in the testes DNA from H-DNA mice on the CD compared to B-DNA mice on the same diet. A significant increase in obesity-induced mutation frequencies in H-DNA vs. B-DNA mice ($p > 0.0001$) demonstrates an obesity-mediated impact on H-DNA-forming sequences in mouse testes.

We observed a significant increase in point mutation frequencies ($p > 0.0001$) in testes tissue DNA from B-DNA mice on the HFD compared to those on the CD (Fig. 2h, Supplementary Data 3). A modest but significant increase ($p = 0.0180$) in point mutations was also observed in H-DNA mice on the HFD compared to the CD. However, we did not find a significant increase in point mutations when comparing H-DNA to B-DNA mice on the CD ($p = 0.3783$) or on the HFD ($p = 0.3730$).

Regarding the frequencies of large deletions in testicular DNA, while we observed a significant increase ($p > 0.0001$) in H-DNA mice on the HFD compared to the CD, this difference was not significant in B-DNA mice ($p = 0.2005$) (Fig. 2i, Supplementary Data 3). A modest increase ($p = 0.0015$) in large deletion frequencies was observed in testes DNA from the H-DNA mice on the CD compared to B-DNA mice on the same diet. However, a marked increase ($p > 0.0001$) in point mutations was observed in obese H-DNA mice compared to obese B-DNA mice. Overall, the results suggest that the mutation profile of testicular tissue DNA from obese B-DNA mice is predominantly characterized by point mutations. In contrast, the combined effect of obesity and H-DNA-induced mutagenesis in the testicular tissue is more inclined toward large deletions. Interestingly, in obese H-DNA mice, ~18% of the large deletion mutants exhibited rearrangements around the H-DNA sequence. The presence of 35-bp insertions aligning with the mouse bone morphogenetic protein 4 (BMP-4) gene sequence provides evidence of repetitive DNA sequence-mediated genomic instability.

## Deletion mutations mapped to the H-DNA-forming region

As part of the mutation spectra analyses, we characterized the length and types of microhomologies at the large deletion junctions in mutant sequences in genomic DNA from B-DNA and H-DNA mice on the CD and HFD diets. In the liver tissue DNA from the B-DNA mice on the CD, ~24% of the deletion junctions were joined with the sequence TAT, whereas ~18% were joined with a GT in mice on the HFD (Fig. 3a). Most of these microhomologies occurred upstream of the *lacZ* mutation-reporter gene, indicating its deletion. In the H-DNA mice, the

deletion junctions joined with AG, G, and GTCAT microhomologies in both diet groups. In obese H-DNA mice, we found G to A, C to T, T to C transitions, and C to G transversions, indicating complex processing of the H-DNA-induced DSBs. In some of these mice, longer (1–16 bp) sequences were detected at the deletion junctions suggesting extensive processing of the DNA ends before microhomology-mediated end-joining.

In the brain tissue DNA, ~28% of the deletion breakpoints from the B-DNA mice on the CD joined with the sequence AGT (Fig. 3b). Whereas 81–90% of the deletion breakpoints in the H-DNA mice on either diet joined with GTCAT microhomologies, suggesting that this could be an outcome related to H-DNA structure-mediated processing, but further investigation is warranted. Interestingly, the remaining 10-19% of deletion junctions in the H-DNA mice on either diet showed 1-10 bp unique sequences. We also noted C to T, T to C transitions, and C to G, G to A transversions enabling the end-joining of DSBs in deletion mutants from the H-DNA mice on the HFD. Similar to the liver tissue, in B-DNA mice on either diet, some microhomologies occurred upstream of the *lacZ* gene, indicating its deletion. In contrast, deletion events frequently occurred within the *lacZ* gene and its promoter region in the H-DNA mice on both diets. These results are consistent with the mutation frequency data observed in the H-DNA mice on the HFD providing further evidence that metabolic changes related to obesity exacerbated H-DNA-mediated genomic instability in the brain.

We found that AG and G microhomologies in the testes tissue were common to the control B-DNA mice on either diet (Fig. 3c). However, in the H-DNA mice on either diet, 16–35% of the deletion junctions contained AG microhomologies, and a GT sequence joined another 20% of them. Moreover, C to T transitions and C to G transversions were found in several microhomology-mediated breakpoint junctions in the obese H-DNA mice. Unlike the liver, we found longer (1-16 bp) microhomologies exclusively in testes DNA from obese B-DNA mice, suggesting extensive end processing before joining. This also indicates that obesity alone could influence the modulation of DNA repair enzymes participating in the complex processing of ends in DSB repair.

Quantification of small (1–5 bp) and large (6–16 bp) microhomologies observed at the deletion junctions recovered from the liver, brain, and testes tissues from B-DNA and H-DNA mice is shown in Fig. 3d. Combining the analyses of mutation frequencies and mutation spectra reveals that H-DNA is inherently more mutagenic in the liver, brain, and testes tissues relative to B-DNA. Additionally, though mutagenesis was only modestly increased by obesity in control B-DNA, it was significantly enhanced by the presence of H-DNA in a tissue-specific manner.

## Obesity exacerbates mutations in H-DNA: insights from NGS analysis

In addition to the *lacZ*-based mutation assays, we utilized Illumina MiSeq Next Generation Sequencing (NGS) on genomic DNA from mouse liver tissue as an orthogonal assay to assess mutations. This deep sequencing technology allowed us to quantify mutations within a 446-bp amplicon spanning the 686–1132 bp region of the p2RT mutation reporter encompassing the H-DNA-forming sequence. Alignment of approximately 15 million NGS reads with the reference p2RT mutation-reporter sequence revealed a marked increase in percent non-reference alleles (comprising point mutations, insertions, and deletions) in and around the H-DNA-forming region. The increase was notably pronounced in the H-DNA mouse subjected to the HFD compared to the mouse on the CD, as shown in Fig. 3e. The overall average percent non-reference alleles across the amplicon were 0.0094 in the mouse on the HFD contrasting with 0.0039 in the mouse on the CD. Within the H-DNA-forming region (253–276 bp of the amplicon) the mean percent non-reference alleles were markedly increased in the mouse on the HFD (0.0197) compared to the mouse

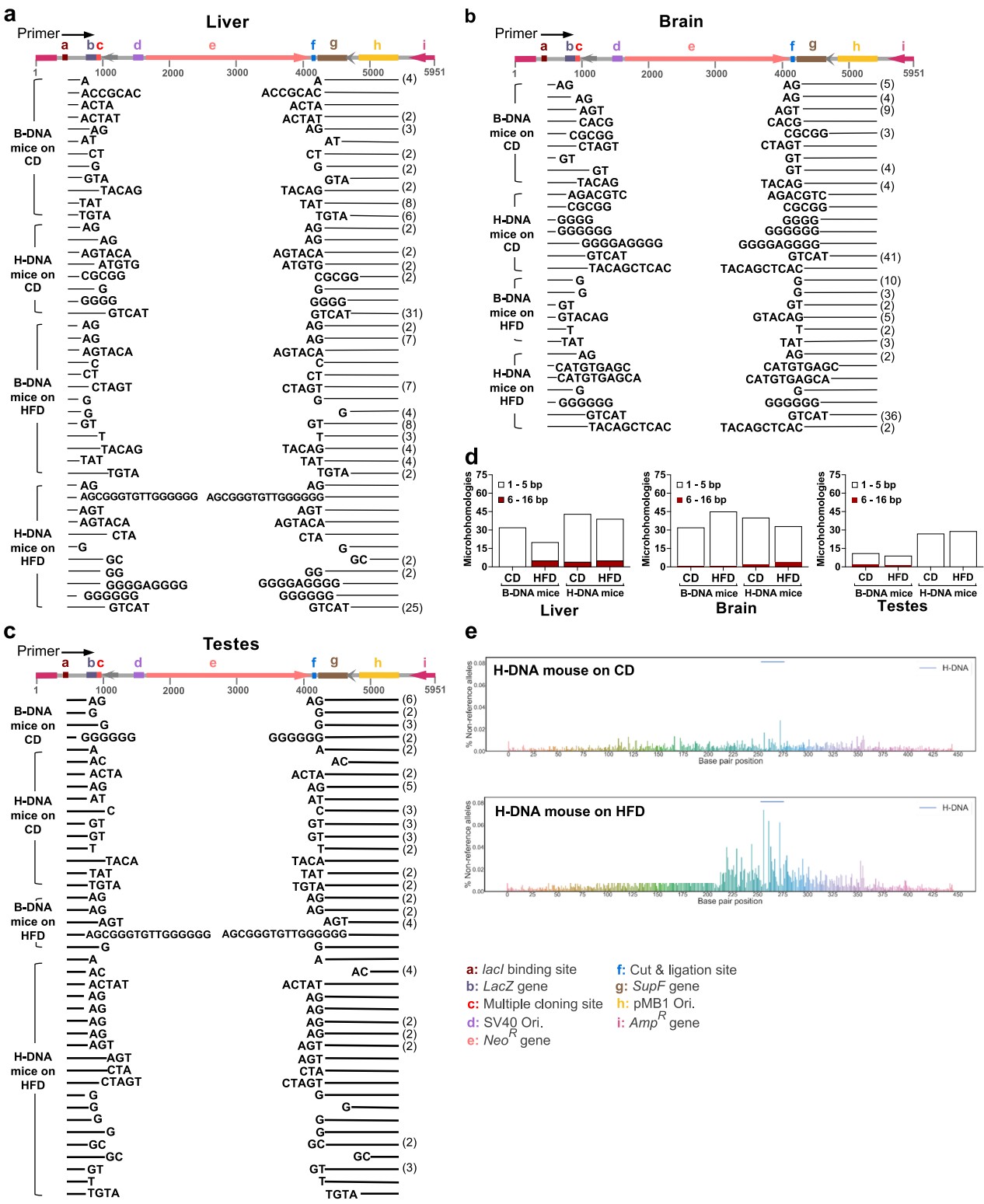

**Fig. 3 | Mutations mapped to the H-DNA-forming region.** Deletion mutations were mapped for their precise position and size by examining the reporter DNA recovered from **a** liver ($n = 157$), **b** brain ($n = 148$), and **c** testes ($n = 79$) tissues from B-DNA mice (CD, $N = 5$; HFD, $N = 5$) and H-DNA mice (CD, $N = 4$; HFD, $N = 5$). The linearized p2RT mutation-reporter sequence (map illustrated in color) served as the reference map. The blank regions between the lines indicate deletions, and the bases at the ends of the lines represent microhomologies at the deletion junctions. Each deletion junction typically features a single copy of microhomology but is listed on both ends since as it cannot be assigned to either side. The number of identical deletion mutants is denoted in parenthesis to the right. **d** Quantification (represented as superimposed bars) of small (1–5 bp) and large (6–16 bp) micro-homologies observed at deletion junctions in mutation reporters recovered from liver, brain, and testes tissues from B-DNA and H-DNA mice. **e** Illumina Mi-Seq NGS deep sequencing across a 446-bp mutation reporter amplicon derived from genomic DNA from liver tissue of an H-DNA mouse on the CD and the HFD shows percent non-reference alleles. The blue bar represents the H-DNA-forming sequence region (253–276 bp) within the amplicon. The 446-bp amplicon corresponds to the 686-1132 bp region of the p2RT mutation reporter containing the H-DNA-forming sequence. Source data are provided as a Source Data file.

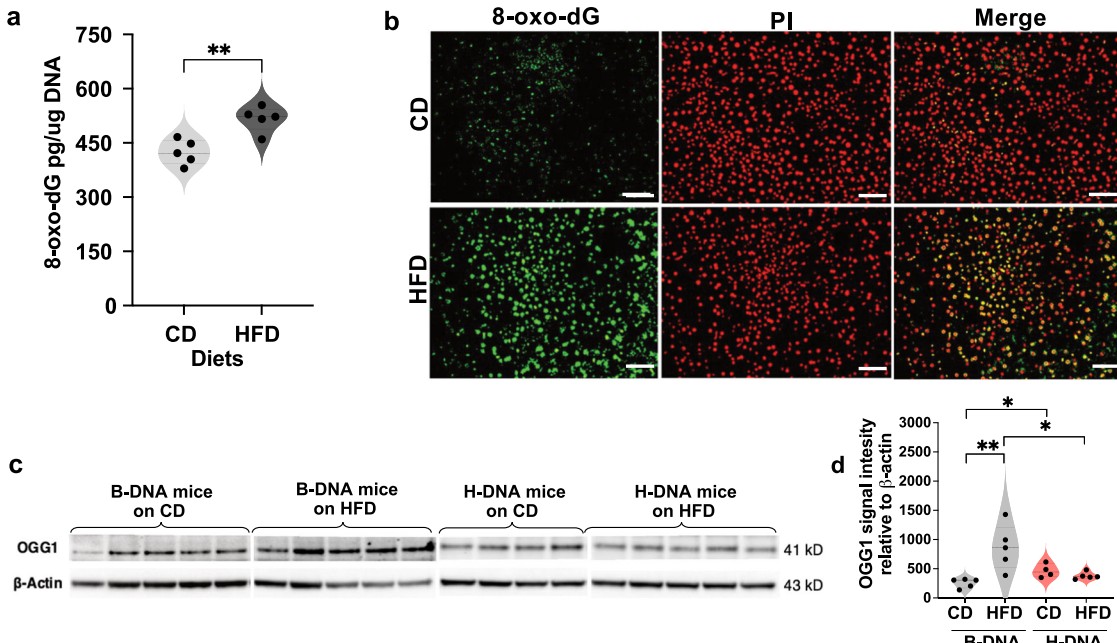

**Fig. 4 | Obesity increases oxidative DNA damage. a** Quantitation of 8-oxo-dG by immunoassay in genomic DNA from B-DNA mice (CD, $N = 5$; HFD, $N = 5$) is represented as a violin plot with all data points and dotted lines indicating median and quartile. Statistical analysis was performed using unpaired, two-tailed Student's $t$ test ($t(8) = 4.229$, $**p = 0.0029$), with 95% confidence interval. **b** Representative images of double immunofluorescence staining for 8-oxo-dG (green) and nucleus (red, PI) in liver tissue sections from mice on the CD and the HFD. Scale bar 70.5 mm. The images were uniformly contrast-enhanced for clarity. **c** Western blot analysis of OGG1 protein (normalized to the loading control β-actin) from testicular tissue extracts. For comparison, protein lysates from B-DNA mice (CD, $N = 5$; HFD, $N = 5$) were run on one blot and H-DNA mice (CD, $N = 4$; HFD, $N = 5$) were run on a separate blot. **d** Quantitation of OGG1 protein represented as a violin plot with all data points and dotted lines indicating median and quartile. Statistical analysis was performed using unpaired, two-tailed Mann–Whitney $U$ test with 95% confidence interval. Diet factor (CD vs HFD): B-DNA ($**p = 0.0079$), H-DNA (not significant, $p = 0.2857$). B-DNA vs. H-DNA factor: CD ($*p = 0.0159$), HFD ($*p = 0.0159$). Source data are provided as a Source Data file.

on the CD (0.0060). These findings were corroborated by replicating the analysis in two additional H-DNA mice from each diet group (Supplementary Fig. 2). In these mice, the mean percent non-reference alleles across the amplicon were 0.0076 in mice on the HFD versus 0.0048 in mice on the CD. In the H-DNA-forming region, this value was markedly elevated in the mice on the HFD (0.0197) compared to the mice on the CD (0.0057). These NGS-derived results also validate our mutation-reporter findings, indicating that obesity increases the mutagenic potential of this endogenous mutation hotspot in vivo.

### Obesity increases oxidative DNA damage in mice

We sought to explore an additional factor, 8-oxo-dG lesions, that could contribute to obesity-mediated genomic instability, in addition to the presence of endogenous H-DNA-forming sequences. Performing the blue-white mutagenesis assay on tissues isolated from obese mice posed technical challenges due to their high lipid content. Consequently, brain and testes tissue were exclusively flash frozen, while the larger liver size allowed for a combination of flash freezing and formalin fixation for mutation screening, and immuno-staining. To assess levels of 8-oxo-dG in genomic DNA, liver tissues from mice on the CD and the HFD were subjected to immunoassay and immunostaining techniques. The immunoassay results revealed a significant increase ($p = 0.0029$) in the oxidative DNA damage marker levels, 8-oxo-dG, in mice on the HFD compared to those on the CD (Fig. 4a). Furthermore, immunostaining of the liver tissue corroborated these findings, as illustrated in Fig. 4b.

In humans, 8-oxo-dG lesions are typically repaired via the base-excision repair (BER) pathway. The primary enzyme responsible for the initial recognition and excision of this lesion is 8-oxo guanine glycosylase (OGG1)[59]. In response to increased oxidative DNA damage, we observed a significant increase ($p = 0.0079$) in the levels of OGG1 protein in B-DNA mice on the HFD compared to those on the CD

(Fig. 4c, d). A similar result was not observed in the H-DNA mice on HFD compared to those on CD, perhaps due to the significantly elevated ($p > 0.05$) OGG1 level in H-DNA mice regardless of the diet.

### Obesity exacerbates the formation of H-DNA-induced SSBs and DSBs in mice

To investigate the mechanisms behind the elevated mutation frequencies and diverse mutation spectra observed in the H-DNA mice, we utilized a DNA modification landscape assay[60] to measure SSBs within and surrounding the H-DNA region in the brain tissue. This assay generates a DNA damage index (DDI) profile across the region of interest, allowing for the sensitive detection and quantification of DNA damage. Our analysis of DDI across the target B-DNA or H-DNA region (Fig. 5a) revealed that while there was no difference in HFD-induced DDI in B-DNA mice ($p = 0.0557$) compared to those on the CD, the difference was significant in H-DNA mice ($p = 0.0007$) when compared to their counterparts on the CD. Interestingly, we did not find a significant increase ($p = 0.7839$) in H-DNA-mediated DDI alone when compared to B-DNA mice on the CD. However, obesity significantly increased ($p > 0.0001$) the DDI in H-DNA mice compared to B-DNA mice. Our results suggest that the unique structural vulnerabilities of H-DNA may result in increased DNA damage, leading to SSBs in the obese state.

Interestingly, Ng et al. reported that somatic and meiotic DSBs tend to localize near the *MYC* oncogene[61]. In our study, we observed increased frequencies of deletions within the H-DNA-forming sequences from the *c-MYC* gene in DNA isolated from the liver, brain, and testes tissues of obese mice. To gain further insights into the identification of DSBs within and surrounding the H-DNA-forming sequence, we employed ligation-mediated PCR (LM-PCR) on mouse genomic DNA (Fig. 5b). We identified two DSB hotspots (at 250 bp and 300 bp) near the H-DNA sequence in DNA extracted from the liver

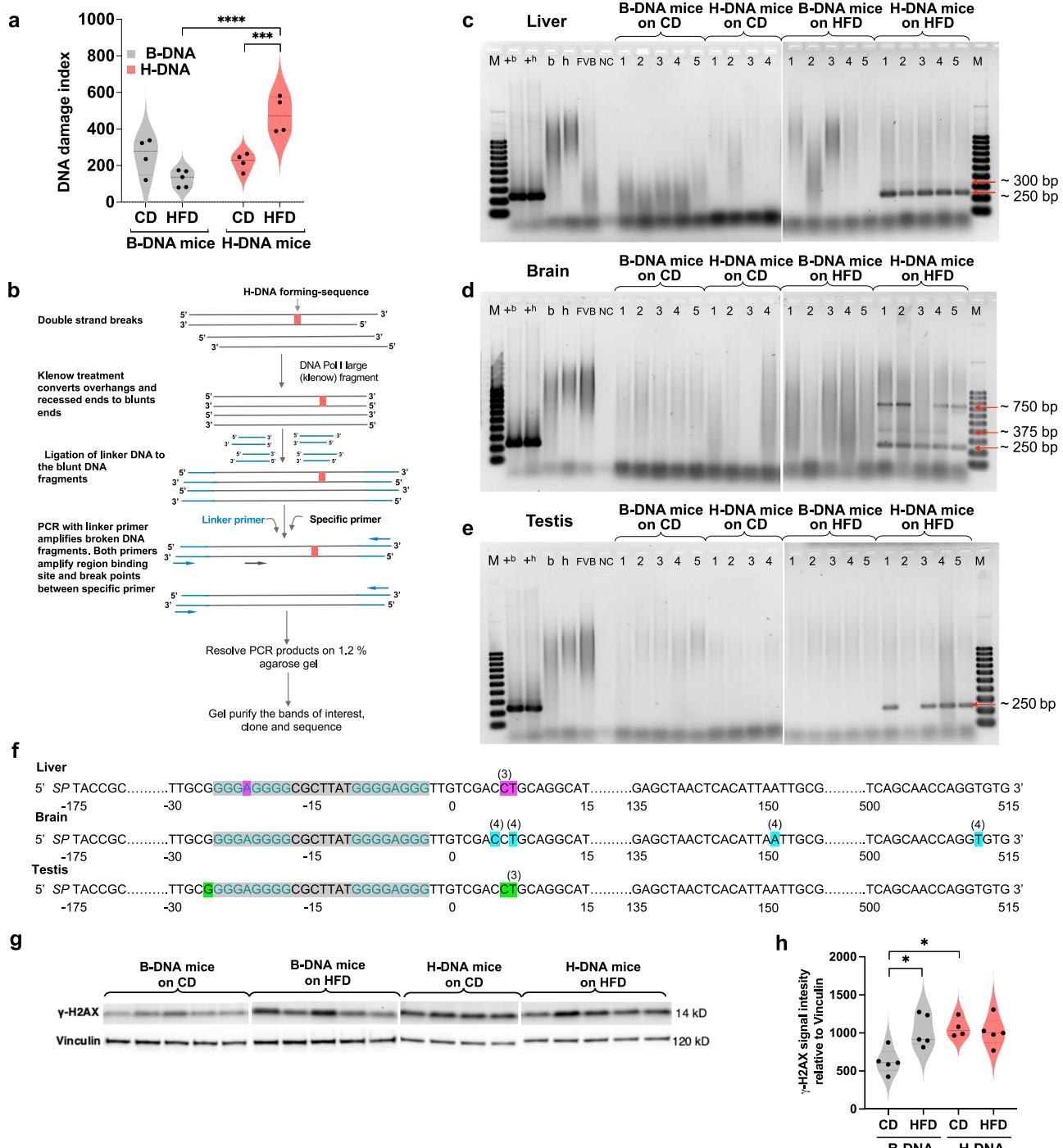

**Fig. 5 | Obesity increases H-DNA-induced SSBs and DSBs. a** The abundance of SSBs measured as DNA damage index in brain DNA from B-DNA mice (CD, $N = 4$; HFD, $N = 5$), and H-DNA mice (CD, $N = 4$; HFD $N = 4$) is represented as a violin plot with all data points and dotted lines indicating median and quartile. Statistical analysis was performed using two-way ANOVA for significance of interaction (diet x B-DNA/H-DNA): $F_{(1,13)} = 26.77$, ***$p = 0.0002$ followed by Sidak multiple comparison test for diet factor (CD vs. HFD): $F_{(1,13)} = 3.157$, not significant, $p = 0.0990$ [B-DNA: not significant, $p = 0.0557$, H-DNA: ***$p = 0.0007$]; and B-DNA vs. H-DNA factor: $F_{(1,13)} = 18.06$, ***$p = 0.0009$ [CD: not significant, $p = 0.7839$, HFD: ****, $p > 0.0001$]. Each p value is adjusted to account for multiple comparisons with family-wise alpha threshold and confidence interval of 95%. **b** Schematic outline of the mapping of DSBs using LM-PCR. **c-e** DSBs were mapped in DNA isolated from the liver, brain, and testes tissues of B-DNA mice (CD, $N = 5$; HFD, $N = 5$), and H-DNA mice (CD, $N = 4$; HFD, $N = 5$). Different lengths of DSB hotspots relative to the H-DNA-forming sequences are shown as red arrows. M, 100 bp DNA ladder; +b, positive control for B-DNA; +h, positive control for

H-DNA; b, B-DNA-containing reporter; h, H-DNA-containing reporter; FVB, FVB mouse genomic DNA; NC, negative control for PCR. **f** The position of DSB breakpoint hotspots in genomic DNA from H-DNA mice on HFD is shown relative to the H-DNA-forming sequence. The number of identical breakpoints in sequences analyzed for the liver ($n = 4$, pink), brain ($n = 10$, blue), and testes ($n = 4$, green) is noted in parenthesis. **g** Western blot analysis of γH2AX protein (normalized to the loading control Vinculin) from testicular tissue extracts of B-DNA mice (CD, $N = 5$; HFD, $N = 5$) and H-DNA mice (CD, $N = 4$; HFD, $N = 5$). For comparison, protein lysates from B-DNA mice (CD, $N = 5$; HFD, $N = 5$) were run on one blot and H-DNA mice (CD, $N = 4$; HFD, $N = 5$) were run on a separate blot. **h** Quantitation of γH2AX protein represented as a violin plot with all data points and dotted lines indicating median and quartile. Statistical analysis was performed using unpaired, two-tailed Mann–Whitney $U$ test with 95% confidence interval. Diet factor (CD vs. HFD): B-DNA (**$p = 0.0159$), H-DNA (not significant, $p = 9048$). B-DNA vs. H-DNA factor: CD (*$p = 0.0159$), HFD (not significant, $p = 0.8413$). Source data are provided as a Source Data file.

tissue of obese H-DNA mice (Fig. 5c). Similarly, we detected three distinct DSB hotspots in the brain tissue DNA (at approximately 250 bp, 375 bp, and 750 bp) (Fig. 5d) and the testes tissue DNA (at approximately 250 bp) (Fig. 5e) from H-DNA mice on the HFD. We acknowledge the sensitivity limitation of this assay, as the DSBs hotspots i.e., accumulation of DSBs within a particular region were exclusively detected in obese H-DNA mice and not in obese B-DNA mice. However, these results align with the increased frequency of large deletions observed in the obese H-DNA mice compared to B-DNA mice. Sequence analysis confirmed that the DSBs occurred within and around the H-DNA-forming sequence (Fig. 5f). These findings support our hypothesis that metabolic changes in the obese state enhance H-DNA-induced SSBs and DSBs, resulting in increased genomic instability at the endogenous mutation hotspots in vivo.

As a cellular response to the induction of DNA damage, we also measured protein levels of the DSB marker γ-H2AX[62] in the testicular tissue of B-DNA and H-DNA mice. We observed a significant increase ($p = 0.0159$) in the levels of γ-H2AX protein in B-DNA mice on the HFD compared to those on the CD (Fig. 5g, h). In contrast, the difference between the CD and the HFD groups of H-DNA mice was not statistically significant, partially due to the increased γ-H2AX levels in H-DNA mice on even a CD. Nevertheless, the results suggest that the H-DNA-forming sequences and the obesity-inducing HFD diet could increase DNA breakage in vivo.

### Obesity reduces the end-joining repair efficiency of DSBs in mice

To evaluate the effect of obesity on the efficiency of DSB end-joining repair, we utilized a fluorophore-based assay developed recently in the lab (Kompella, del Mundo & Vasquez; unpublished) (Fig. 6a). Fluorophore-labeled duplexes mimicking various DSB ends were incubated with testicular tissue extracts from mice on either a CD or HFD, resulting in the formation of end-ligation products. Specifically, we investigated 5'-5' compatible ends that can be ligated directly (Fig. 6b) and 5'-5' non-compatible ends that must be processed before ligation (Fig. 6c). Statistical analysis was performed to determine differences in repair efficiency, measured as the joining of two duplexes to form shorter-length (100–150 bp) ligation products or the joining of more than two duplexes to form longer-length (450–500 bp) ligation products.

It is worth noting that the cell-free extracts from tissues contain active nucleases and other DNA repair/processing proteins, which may contribute to the degradation of both un-ligated substrates and ligated products. Although unavoidable, this is an indicator of the activity and functionality of cell-free tissue extracts. Therefore, we utilized testicular tissue extract for this assay, known for its high DSB end-joining efficiency and moderate nuclease activity compared to other tissues[63,64]. During assay development, we observed that the kinetics of compatible end-joining was remarkably faster within the initial hour of substrate incubation in the tissue extract. However, prolonged incubation (>4 h) to promote the formation of longer products resulted in DNA loss. Conversely, for substrates with non-compatible ends requiring processing, the kinetics appeared slower, with maximum multimerization occurring by 4 h and further incubation resulting in DNA loss. To ensure consistency, both compatible and non-compatible end substrates were incubated for the same duration of time (4 h), facilitating a fair comparison between the two conditions.

As depicted in Fig. 6d, e, DNA substrates with compatible ends incubated in testicular tissue extract from obese mice resulted in an -8.4% decrease in shorter-length ligation products compared to the mice on the CD ($t(7) = 4.405$, $p = 0.0031$). However, the formation of longer-length ligation products did not differ significantly between the two diet groups ($t(7) = 0.9588$, $p = 0.3696$) (Fig. 6f). Conversely, DNA substrates with non-compatible ends, which require end processing before ligation, showed similar levels of shorter-length ligation

product formation ($t(8) = 0.5954$, $p = 0.5681$) when incubated in testicular tissue extract from mice on the HFD compared to the CD. (Fig. 6g). However, the two diet groups significantly differed in the ability to form longer-length ligation products. Typically, DSBs with non-compatible ends undergo processing to generate compatible or blunt ends before ligation. Interestingly, we observed an -12.8% reduction ($t(8) = 2.394$, $p = 0.0436$) in the formation of longer-length ligation products when DNA substrates with non-compatible ends were incubated with testicular tissue extracts from obese mice compared to extracts from normal-weight mice (Fig. 6h). These findings suggest that in the obese state, DNA end-processing of such intermediates may be compromised (Fig. 6i), resulting in DSBs, error-generating repair, and subsequent genomic instability.

### Obesity alters the expression of DSB end-joining repair proteins in mice

We examined the levels of proteins involved in the canonical NHEJ repair pathway in testicular tissue extracts from B-DNA and H-DNA mice on the diets via immunoblotting (Fig. 7a, b). We observed a significant reduction in Ku70 protein levels in both B-DNA and H-DNA mice on the HFD compared to their counterparts on the CD, indicating impaired recognition of DSBs in the obese state. Interestingly, overall Ku70 protein levels were significantly elevated in H-DNA mice compared to the B-DNA mice on either diet providing further evidence for the increased occurrence of H-DNA-mediated DSBs observed earlier. Following DSB ends stabilization, the catalytic subunit of DNA-dependent protein kinase (DNA-PKcs) is recruited to form the active DNA-dependent protein kinase (DNA-PK). We did not observe a significant difference in the levels of the 230 kD subunit of DNA-PK between B-DNA mice on either diet. However, DNA-PK was significantly reduced in H-DNA mice on the HFD compared to those on the CD and B-DNA mice on the HFD. This suggests a possible impairment in DSB recognition and binding to DNA ends, leading to the accumulation of DSBs, as observed in HFD-fed H-DNA mice. Interestingly, X-Ray Repair Cross Complementing 4 (XRCC4) levels were notably increased in B-DNA mice on the HFD compared to those on the CD and H-DNA mice on either diet. However, the Ligase IV levels remained consistent across both B-DNA and H-DNA mice on either diet.

Microhomologies at the deletion junctions of DSBs led us to speculate the involvement of microhomology-mediated end-joining (MMEJ) pathway proteins, as shown in (Fig. 7a, c). We found that the MRE11 protein, a component of the MRN complex - Meiotic recombination 11 (MRE11), RAD50, and NBS1 involved in DSB end resection, was found to be significantly increased in obese B-DNA mice and in H-DNA mice on either diet. Interestingly, RAD50 levels were markedly increased in obese B-DNA mice compared to those on a CD, although there was a substantial intra-animal variation. A study investigating the enzymatic machinery of the MMEJ pathway reported that inhibiting RAD50 expression did not reduce MMEJ[65]. X-Ray Repair Cross Completing I (XRCC1) and Ligase III are known to be required for joining DSB ends[66]. The upregulation of XRCC1 and Ligase III in B-DNA mice on the obesity-inducing diet and H-DNA mice on both diets, compared to B-DNA mice on the CD, further suggests that the repair of obesity- and H-DNA-induced DSBs is carried out through an error-prone MMEJ pathway. Further research is warranted to investigate the precise mechanisms of alternate-structure-induced DSB repair in the obese state.

### Discussion

Our study contributes to the growing evidence linking obesity to genomic instability. It also underscores the importance of understanding the molecular alterations induced by obesity on DNA repeat-mediated genomic instability. Using transgenic mutation-reporter mice, we examined the impact of obesity on these cancer-associated endogenous mutation hotspots. Our results revealed that dietary fat elicited obesity and liver steatosis in both control mice (B-DNA) and mice

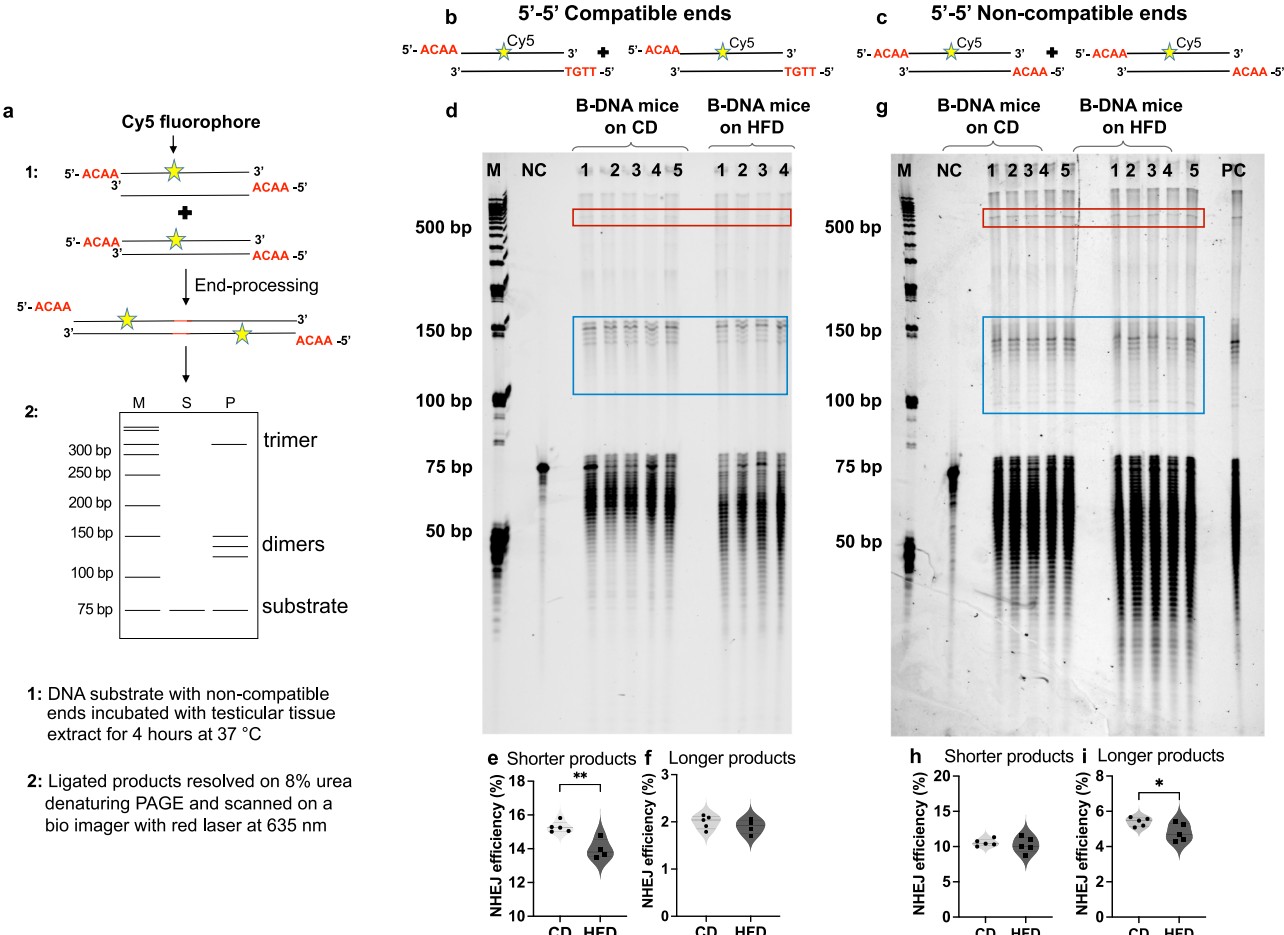

**Fig. 6 | Obesity reduces the end-joining repair efficiency of DSBs. a** Schematic outline of the fluorophore-based assay for joining DSBs with non-compatible ends using mouse testicular tissue extract. **b** Fluorophore-tagged DNA substrate mimicking DSBs with 5′(ACAA)-5′(TGTT) compatible ends. **c** Fluorophore-tagged DNA substrate mimicking DSBs with 5′(ACAA)-5′(ACAA) non-compatible ends. **d** Representative image of denaturing urea polyacrylamide gel showing ligation products of 5′-5′ compatible ends using testicular tissue extract from B-DNA mice (CD, $N=5$; HFD, $N=4$). **e** Quantitation of shorter length ligation products shown in d (blue box, $t(7)=4.405$, **$p=0.0031$). **f** Quantitation of longer ligation products shown in d (red box, $t(7)=0.9588$, not significant, $p=0.3696$). **g** Representative image of denaturing urea polyacrylamide gel showing ligation products of 5′-5′ non-compatible ends using testicular tissue extract from B-DNA mice (CD, $N=5$; HFD, $N=5$). **h** Quantitation of shorter length ligation products shown in g (blue box, $t(8)=0.5954$, not significant, $p=0.5681$). **i** Quantitation of longer ligation products shown in g (red box, $t(8)=2.394$, *$p=0.0436$). Data for **e**, **f** and **h**, **i** is represented as violin plots with all data points and dotted lines indicating median and quartile. Statistical analysis was performed using unpaired, two-tailed student's t-test with 95% confidence interval. Source data are provided as a Source Data file. M 50 bp DNA ladder, NC negative control, PC positive control.

containing a human DNA sequence capable of forming a mutagenic alternative DNA structure (H-DNA). Notably, the mutation frequencies in genomic DNA from the liver, brain, and testes tissues were significantly affected by the obesity-inducing diet in both B-DNA and H-DNA mice, with a more pronounced effect observed in H-DNA mice. These findings indicate that obesity independently contributes to genomic instability, potentially through mechanisms related to altered metabolic processes that could lead to increased DNA damage and inefficient DNA repair[67]. Additionally, H-DNA mice on the CD exhibited amplified mutation frequencies compared to the control B-DNA mice emphasizing the inherent mutagenic properties of H-DNA-forming sequences in vivo. Furthermore, we observed a higher frequency of large deletions and DSBs in H-DNA mice over that in B-DNA mice, suggesting the greater vulnerability of these sequences to genomic instability.

Tissue-specific variations in mutation frequencies may be influenced by differences in cell turnover rates. Frequent cell division increases the chances of errors occurring during DNA replication, leading to a higher mutation rate[68,69]. We observed an intriguing pattern where the brain showed the highest accumulation of mutations despite having the lowest cell turnover rate[70,71]. Our findings underscore that under conditions of metabolic stress, such as obesity, the brain may accumulate mutations at endogenous mutation hotspots beyond its repair capacity. This observation is particularly relevant for non-proliferating cells, such as the post-mitotic neurons, where damaged DNA can hinder the progression of RNA polymerase during transcription, leading to the accumulation of DNA damage[72,73]. We also found that obesity-related factors could contribute to mutation accumulation in testicular cells, although to a lesser extent than brain and liver tissues. This is likely because testes have evolved mechanisms to prioritize a low cell turnover rate to maintain the long-term fidelity and accuracy of cell division and gamete propagation[74]. Conversely, the liver tissue has a significant capacity for cell turnover[75] and responds to cellular stress and damage caused by lipid accumulation by adapting its cell turnover and metabolic processes. However, chronic low-grade inflammation, oxidative stress, and metabolic dysfunction associated with obesity can potentially contribute to DNA damage and increase the risk of mutations over time, especially in hotspots of genomic instability (e.g. H-DNA). In future studies, we aim to explore how obesity may differentially affect DNA repeat-mediated genomic stability in hepatocytes and immune cells within the liver, potentially contributing to the development of liver dysfunction, including hepatocellular carcinoma.

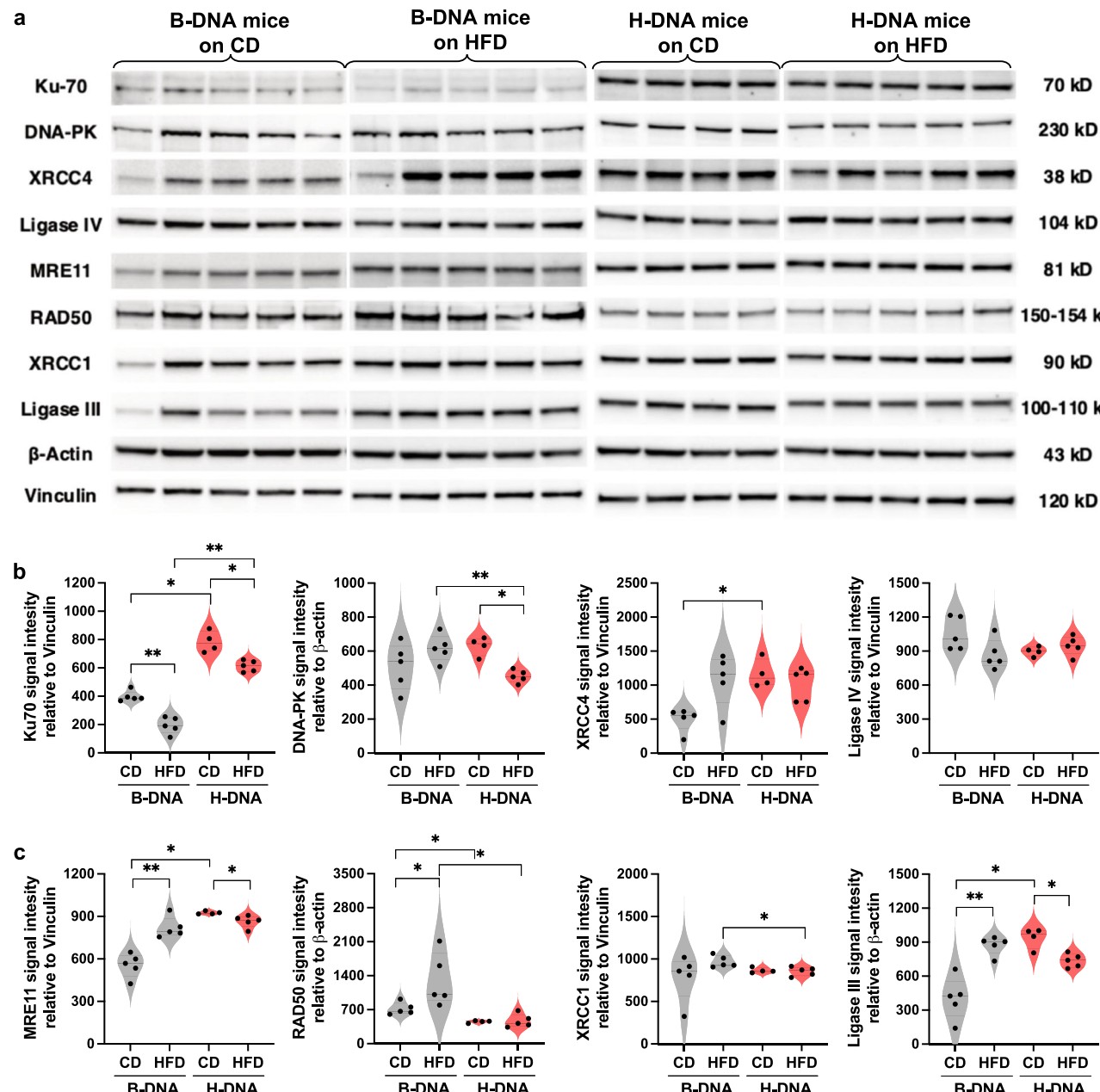

**Fig. 7 | Obesity alters the levels of DSB end-joining repair proteins. a** Western blot analysis of NHEJ pathway proteins: Ku70, DNA-PK, XRCC4, Ligase IV, and alternative-NHEJ pathway proteins: MRE11, Rad50, XRCC1, Ligase III (normalized to loading controls β-actin or Vinculin) from testicular tissue extracts. For comparison, protein lysates from B-DNA mice (CD, N = 5; HFD, N = 5) were run on one blot and H-DNA mice (CD, N = 4; HFD, N = 5) were run on a separate blot. **b, c** Quantitation of blots represented as violin plots with all data points and dotted lines indicating median and quartile. Statistical analysis was performed using unpaired, two-tailed Mann–Whitney $U$ test with 95% confidence interval. $p > 0.05$ (not significant), *$p > 0.05$, **$p > 0.01$. Source data are provided as a Source Data file. Statistical analysis is provided as Supplementary Note 2.

Our findings also indicate that H-DNA-forming sequences (from a translocation breakpoint hotspot in the *c-MYC* gene in Burkitt lymphoma) confer inherent mutagenic properties regardless of the tissue type. This is consistent with our previous findings of replication-dependent and replication-independent mechanisms of H-DNA-induced mutagenesis[54]. These sequences represent mutation hotspots in mammalian genomes, and under the influence of obesity-induced metabolic stress, the mutagenic potential of H-DNA-forming sequences is further exacerbated. These insights highlight the importance of considering tissue-specific factors and the inherent characteristics of DNA sequences when assessing the mutational landscape associated with obesity-related genomic instability.

This study's findings can be integrated into different human cancer risks associated to obesity. For example, it is worth noting that several alternative DNA structure-forming sequences have been implicated in various neurological disorders[17–19,76]. Although we did not specifically examine different brain regions, exploring the association between H-DNA-forming sequences and neurological disorders such as Alzheimer's disease (AD) and dementia using NGS is warranted, considering recent reports linking obesity with these conditions[77]. Understanding the potential involvement of H-DNA in such neurological disorders could provide valuable insights into their pathogenesis and open new avenues for their early diagnosis and treatment. Additionally, investigating the *c-MYC* oncogene region, which co-localizes with alternate DNA structure-forming sequences in obese patients

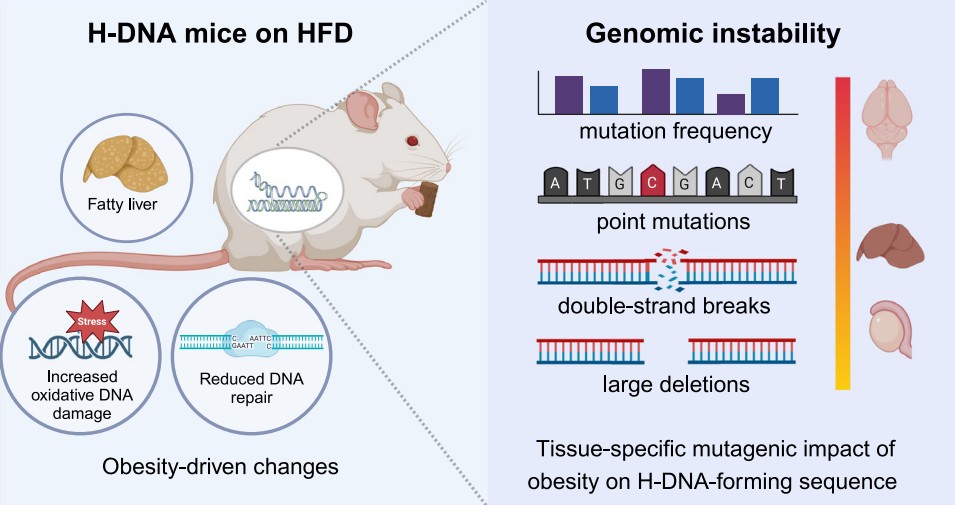

**Fig. 8 | Obesity increases DNA repeat-mediated genomic instability.** Obesity stimulates liver steatosis and oxidative stress. Obesity amplifies the mutagenic potential of H-DNA in mouse liver, brain, and testes tissues. Point mutations, DSBs, and large deletions can be mapped specifically to the H-DNA-forming region. DNA damage repair capacity is altered in the obese state, further contributing to genomic instability. This figure was created with BioRender.com and released under a Creative Commons Attribution-NonCommercial-NoDerivs 4.0 International license.

could shed light on the increased risk of developing translocation-related leukemia and lymphoma.

Clinical studies have shown urinary 8-oxo-dG levels as a useful predictor of oxidative stress-mediated genomic instability during obesity[78,79]. We observed a significant increase in 8-oxo-dG levels in obese mice compared to normal weight mice. Previous studies by the Llyod (2012) and Sampath (2018) groups have demonstrated the susceptibility of the 8-oxo-dG repair enzyme, OGG1-deficient mice to adiposity and steatosis under HFD conditions[80], while overexpression of mitochondrial OGG1 offers protection against diet-induced obesity[81]. In our study, we observed an upregulation of OGG1, likely a protective, albeit insufficient, response aimed at mitigating obesity-induced oxidative DNA damage. Remarkably, these results are consistent with a recent study reporting upregulation of the OGG1 gene in the visceral adipose tissue of individuals with obesity[82].

Oxidized DNA can result in SSBs via direct[83] and indirect mechanisms[84]. Indeed, the increase in SSBs was significant in obese H-DNA mice compared to the mice with B-DNA-forming-sequence in their chromosomes. Additionally, we have previously demonstrated that H-DNA can stimulate the formation of DSBs via error-generating processing in eukaryotic genomes[54]. Our study provides evidence of obesity-induced stimulation of DSBs at an endogenous mutation hotspot, and of DNA damage responses, as indicated by increased levels of the DSB marker γH2AX in testicular tissue extracts. These observations are consistent with a recent study showing elevated γH2AX foci in small follicles of ovaries from obese vs. lean mice[85]. We have previously reported the occurrence of DSBs induced by H-DNA structures in normal-weight mice[51]. However, under obese conditions, the DSB levels and, subsequently, γH2AX levels were found to be significantly amplified.

Increased DNA damage and inefficient DSB repair could lead to the stimulation of mutations and DSB formation, as shown in the mutagenesis and LM-PCR data in this study. Previous studies have shown that DSBs associated with clustered oxidative DNA lesions are largely repaired by non-homologous end-joining (NHEJ)[35]. We have demonstrated that DSBs induced by H-DNA-forming sequences are repaired through an MMEJ pathway, resulting in large-scale deletions with microhomologies at the breakpoint junctions[51,86]. A recent report also suggests that diet can modulate the efficiency of DNA end-joining repair[87]. For example, short-term calorie restriction can enhance DSB repair in various tissues in mice[88]. While exploring the impact of transitioning from a HFD to a low-fat diet would be intriguing, the comprehensive understanding of this interplay might be influenced by factors such as the age of the mouse at the conclusion of a prolonged study. Moreover, mutations in and around alternative structure-forming regions might be refractory to efficient repair, contributing to mutation accumulation and genomic instability. As expected, our results showed reduced DSB end-joining repair efficiency in tissue extracts from obese mice compared to those from normal-weight mice. While beyond the scope of the present study, future experiments utilizing NHEJ or MMEJ reporter mice[88] engineered to harbor B-DNA- or H-DNA-forming sequences and subjected to dietary interventions could provide comprehensive insights into the dynamic relationship between obesity, age, genomic instability, and cancer risk.

Furthermore, obesity has been found to impact the expression of DNA repair proteins[67] with an observed upregulation of proteins involved in MMEJ repair in our study. Collectively, the findings suggest that the obesity-driven accumulation of DSBs and subsequent genomic instability at endogenous mutation hotspots may result from increased DSBs, decreased repair efficiency, or a combination of both factors. Further investigation into the mechanisms underlying compromised repair efficiency through systematic immunodepletion of DNA repair proteins is warranted.

In conclusion, as depicted in Fig. 8, here we demonstrate molecular alterations induced by obesity at endogenous mutation hotspots within a living system. These findings have broad implications for public health, considering the established link between obesity and increased cancer incidence, progression, and severity, and other genetic disorders[6,89]. By elucidating the role(s) of repeat sequence-mediated structural changes in DNA and their contribution(s) to obesity-related genomic instability, we set the stage for discovering new mechanisms, biomarkers, and therapeutic targets. Continued exploration of these complex interactions will deepen our understanding of the obesity-cancer link and facilitate the development of strategies to prevent, diagnose, and treat obesity-related diseases.

## Methods
### Animal model and diets
The mutation-reporter transgenic mouse model was established according to previously described methods[52]. These mice were engineered to carry a mutation-reporter construct integrated into their chromosomes, consisting of either H-DNA or control B-DNA sequences

located upstream of a *lacZ* mutation-reporter gene. The recoverable p2RT mutation-reporter, spanning 5,951 base pairs (bp), comprises a *lacZ* mutation-reporter gene encoding β-galactosidase, which is susceptible to frameshift mutations, small-scale deletions, large-scale deletions, and expansions in the inserted region[53]. Additionally, the p2RT construct contains replication origins for both mammalian (SV40) and bacterial cells (pMB1), along with two antibiotic-resistance genes (ampicillin and neomycin) for selection. These resistance genes are separated by *lacI*-binding sequences and unique restriction sites (e.g., SpeI), facilitating the recovery of the p2RT mutation reporter from genomic DNA. The p2RT mutation reporter containing either a 29-bp control B-DNA forming sequence (B-DNA mouse) or a 23-bp H-DNA-forming sequence (H-DNA mouse) mapping to a human *c-MYC* translocation breakpoint hotspot in Burkitt lymphoma was integrated into FVB/N mouse chromosomes. For model characterization, the founder mice were genotyped with primers targeting the antibiotic resistance gene and the region encompassing the inserted mutation reporters. Copy numbers of the mutation reporters were estimated by combining quantitative PCR and real-time PCR results, comparing the signal intensity or ΔΔCt of the integrated fragments to that of the *Gapdh* gene. Despite variations in copy number, the mutation frequencies within (B-DNA or H-DNA) and between (B-DNA vs. H-DNA) the mouse type were consistent.

Bodyweight and age-matched male B-DNA and H-DNA mice ($N = 4$–5 per group) were subjected to different dietary interventions for 13 weeks, commencing at 5-6 weeks of age. The normal diet group received a 10 kcal% from fat control diet (CD) [Research Diet D12450B], while the experimental group was fed an obesity-inducing high-fat diet (HFD) containing 60 kcal% from fat [Research Diet D12492][90]. Kcal% from fat is defined as the % of the total calories in that diet coming from fat. General health evaluation was carried out daily, and body weight (g) and food consumed (g) by each mouse were measured weekly for the duration of the study. After the completion of the study, the mice were euthanized using $CO_2$ asphyxiation, and brain, liver, and testes tissues were collected. The tissues were frozen in liquid nitrogen and stored at −80 °C until further analysis.

The University of Texas at Austin Institutional Animal Care and Use Committee (IACUC) protocols AUP-2016-00286 and AUP-2019-00258 approved the study. The mice were housed in polycarbonate cages (5 mice/cage) on sterile bedding and provided with nestlets, shepherd-shacks, and sterile water *ad libitum*. The housing room was maintained at 20–22 °C, 60–70% relative humidity, and a 12/12-h light/dark cycle.

### Genomic DNA isolation from mouse tissues
Approximately 50 mg of frozen mouse tissue was finely minced while kept on dry ice and then mixed with 0.6 mL of lysis buffer (5 mM EDTA, 20 mM Tris-HCl pH 8.0, 400 mM NaCl, and 1% SDS) supplemented with 100 µg/mL RNAse A, 1 mg/mL proteinase K. The samples were incubated at 50 °C with rotation to ensure thorough lysis and release of cellular components. Following overnight incubation, the extract was purified twice with an equal volume of phenol-chloroform-isoamyl alcohol. DNA was ethanol precipitated (0.1 volume of 3 M sodium acetate pH 5.2 and 2.5 volumes of 100% ethanol) and incubated overnight at −20 °C. The following day, after centrifugation for 30 min at $16,000 \times g$ at 4 °C, followed by 70% ethanol wash and air drying, the DNA pellet was resuspended in 100 µL of 10 mM Tris-HCl (pH 8.0) and stored at −20 °C until further use.

### Mutation-reporter DNA recovery
The mutation-reporter DNA was recovered from the genomic DNA following previously described protocols with some modifications[53,91]. The preparation of the lacI-lacZ fusion protein and its coupling with sheep anti-mouse IgG magnetic Dynabeads (Thermo Fisher Scientific, Baltics UAB) was performed according to established methods[91–93]. A minor magnetic bead preparation protocol adjustment involved washing and resuspending the lacI-lacZ fusion protein-coupled Dynabeads in 1 mL of PBS containing 0.1% BSA.

To recover the mutation reporter, 70−80 µg of genomic DNA was digested with 70-80 U of SpeI-HF restriction enzyme (NEB, Ipswich, MA). The digestion was done in a 400 µL reaction volume at 37 °C with rotation overnight and an additional 4 h during the second digestion step. Each digestion was followed by purified by phenol-chloroform extraction and ethanol precipitation, performed overnight at −20 °C. To facilitate the binding of the lacI-lacZ fusion protein to the two *lacI* binding sites in the reporter DNA, the digested genomic DNA was dissolved in 96 µL of water, mixed with 24 µL of 5x binding buffer (10 mM Tris-HCl pH 6.8, 1 mM EDTA, 10 mM $MgCl_2$, 5% glycerol). Subsequently, 100 µL of lacI-lacZ coated Dynabeads were added, and the mixture was incubated for 1 h at 37 °C with rotation. The magnetic beads were washed thrice with 250 µL of 1x binding buffer to remove the genomic DNA. The reporter DNA was then eluted by incubating the pelleted magnetic beads in 200 µL of an elution buffer containing 100 µL of water, 75 µL of IPTG elution buffer (10 mM Tris-HCl, pH 7.5, 1 mM EDTA, 125 mM NaCl), 20 µL of NEB buffer 2 (NEB, Ipswich, MA), and 5 µL of 25 mg/mL IPTG. The elution step was performed for 1 h at 37 °C with rotation.

The eluted reporter DNA was re-circularized by incubating it with 1 µL of 0.1 U T4 DNA ligase (NEB, Ipswich, MA) and 1 µL of 10 mM ATP for 1 h at room temperature. After pelleting the magnetic beads, the DNA supernatant was subjected to ethanol precipitation in the presence of 20 µg of glycogen overnight at −20 °C. The recovered reporter DNA was re-circularized by incubating the DNA pellet in 10 µL of nuclease-free water and 1x blunt/TA ligase (NEB, Ipswich, MA) for 45 min at 25 °C. This was followed by phenol-chloroform extraction and ethanol precipitation, adding 20 µg of glycogen overnight at −20 °C. Finally, the ligated DNA pellet was resuspended in 8 µL of 10 mM Tris-HCl (pH 8.0) immediately before transformation into bacteria.

### Mutation screening assay and mutant characterization
Mutations in the *lacZ* gene were screened using a blue-white mutagenesis assay, following a previously described protocol, with some modifications[53]. Briefly, 2 µL of the recovered reporter DNA was gently mixed on ice with 25 µL of DH10β *E. coli* electro-competent cells (Thermo Fisher Scientific, Waltham, MA). The transformation was performed according to the manufacturer's instructions using the BioRad GenePulser® II with settings of 2.0 kV, 200 Ω, and 25 µF. The transformed cells were plated on Luria broth (LB) agar plates (Miller formulation) supplemented with X-gal (400 µg/L), carbenicillin (100 µg/L), and IPTG (400 µg/L). The plates were incubated in an inverted position overnight at 37 °C to allow colony formation.

For each tissue (liver, brain, and testes) in each diet group (CD and HFD) of a B-DNA or H-DNA mouse, a minimum of 20,000 blue colonies (representing wild type) were counted. The mutation frequency was determined by calculating the ratio of the total number of white colonies (representing mutants) to the combined number of blue and white colonies counted for each mouse.

Approximately 45−50 mutant colonies for each tissue and diet group of B-DNA or H-DNA mice were streaked on LB agar X-gal plates and incubated overnight at 37 °C. Single white colonies were cultured in 5 mL LB media containing 100 µg/mL carbenicillin overnight at 37 °C. The reporter DNA was retrieved from the bacterial cells using a QIAprep spin miniprep kit. Approximately 300−600 ng/µL of DNA was mixed with 10 pmol of primer 548 (Supplementary Data 4) and submitted for Sanger sequencing. The mutant sequences were analyzed by comparing them with the respective reference sequences using the NCBI Basic Local Alignment Search Tool (BLAST).

For mutation spectrum analysis of B-DNA and H-DNA mice on a specific diet, the total number of mutants with only one dominant

mutation (small deletion, point mutation, or large deletion) was counted for each mouse. Priority was given to large deletions over small deletions, followed by point mutations. The percentage of the dominant mutation was calculated and then multiplied by the corresponding mutation frequency to obtain the total estimated frequency of the mutant type.

### Next-generation sequencing

We designed specific forward and reverse primers (Supplementary Data 4) to amplify a 446-bp region containing the H-DNA-forming sequence within the H-DNA-containing mutation reporter integrated into the mouse genome. The target region was amplified for only 14 cycles with the Kapa HiFi PCR kit (Kapa Biosystems, Wilmington, MA) to minimize PCR-induced mutations. Multiple PCR reactions were combined and purified using SPRIselect paramagnetic beads (Beckman Coulter, Indianapolis, IN). To remove large non-specific products and residual genomic DNA, one volume of the PCR product was mixed with 0.55x volume of SPRIselect beads for 15 min at room temperature. The beads were then pelleted on a magnetic particle concentrator for 10 min, and the supernatant was collected. Primer dimers were eliminated by incubating the supernatant with a 0.8x volume of SPRIselect beads for 15 min at room temperature. After discarding the supernatant, the beads were washed twice with freshly prepared 80% ethanol and air-dried for 7 min (taking care to avoid over-drying). The amplified DNA was eluted by incubating the pelleted beads in 40 μL of 10 mM Tris-HCl pH 8.0 for 15 min at room temperature, followed by overnight ethanol precipitation at −20 °C. The following day, the purified amplicon was pelleted, washed with 70% ethanol, air-dried, and resuspended in 20 μL of 10 mM Tris-HCl pH 8.0. Pre-and post-purification samples were analyzed by electrophoresis on 1% agarose gels and stained with SYBR gold for quality assessment. The purified amplicons were submitted to the University of Texas MD Anderson Cancer Center DNA sequencing facility for NGS using the Illumina MiSeq platform.

For MiSeq sequencing, 5 μL of the control and test libraries were denatured with 0.2 N NaOH for 5 min at room temperature and diluted to 10 pM before loading them onto a MiSeq V3 600 cycle cartridge for cluster generation and sequencing using a paired-end 300 bp run. A 15% PhiX sequencing control library (Illumina, San Diego, CA) was included as a control. After demultiplexing the samples, the millions of overlapping reads obtained from the 300 bp paired-end Illumina sequencing were aligned and merged using the Pandaseq program[94] with the ea_util merging algorithm. The resulting high-quality consensus reads (400−500) was aligned to the reference sequence with bowtie2, converted to bam, and sorted. The bamtools piledriver[95] was utilized to calculate per-base reference and non-reference alleles. The mutation frequency, or alternate allele ratio, was calculated by dividing the total number of alternate sequenced alleles by the total depth of each base. This alternate allele frequency was then visualized across the amplicon reference sequence using a Python-based Jupyter Notebook and matplotlib library.

### Immunostaining of 8-oxo-dG

The harvested liver tissues were fixed in 10% neutral buffered formalin and then transferred to 70% ethanol after 24 h. Paraffin-embedded sections (4 μm) were stained for the 8-oxodG using a double-fluorescent labeling method of immunostaining following a previously described protocol[96]. In brief, the liver sections were subjected to staining with an 8-oxodG antibody (Meridian Life Science, Inc, Memphis, TN). Subsequently, a fluorescein isothiocyanate (FITC)-labeled secondary antibody was used for detection (with excitation maxima ranging from 450−490 nm). The nuclei were stained with propidium iodide (PI, with an excitation maximum of 535 nm). The tissue sections were observed and photographed using a Nikon microscope with epifluorescence capabilities and appropriate excitation and bandpass filters. Control tissue sections were stained without the primary antibody to assess specificity.

### Quantitation of 8-oxo-dG

DNA was extracted from frozen liver tissues following the established protocol mentioned earlier[96]. The purity and concentration of the isolated DNA was determined using a spectrophotometer. Digestion of 8-oxodG was performed using nuclease P1 and alkaline phosphatase as previously described[96]. A standard curve was generated using purified 8-oxodG as a reference to ensure accuracy and reliability. The level of 8-oxo-dG in each sample was quantified with reference to the standard curve using linear regression analyses, following the instructions provided by the manufacturer (Cayman Chemical, Ann Arbor, MI).

### DNA modification landscape assay

Our collaborator, Dr. Yuan Liu at Florida International University (FIU) developed the DNA modification landscape assay[60] used in this study to detect SSBs and DSBs in B-DNA- and H-DNA forming sequences in brain tissue of the B-DNA and H-DNA mice on different diets. Using this assay, SSBs resulting from endogenous oxidative DNA damage 8-oxo-dG, AP sites, nicks, and gaps on the 23-bp H-DNA-forming sequence and a corresponding B-DNA-forming sequence in the brain tissue of B-DNA and H-DNA mice on the different diets were detected. Briefly, 2 μg of genomic DNA was denatured at 95 °C for 5 min resulting in single-stranded DNA. Subsequently, primer extension was performed using a 23-nucleotide primer (primer 1) specifically annealed to the downstream region of the B-DNA and H-DNA forming sequences (Supplementary Data 4) using vent DNA polymerase (NEB, Ipswich, MA). This step generated double-stranded DNA fragments with various sizes that represent different sites of SSBs on the target B-DNA and H-DNA-forming sequences. Next, the DNA fragments were ligated with a barcode DNA and subjected to PCR amplification using primer 2 (annealed to the downstream region closer to B-DNA or H-DNA-forming sequences than primer 1) (Supplementary Data 4) and the barcode primer (IDT, Coralville, IA) (Supplementary Data 4) with the LongAmp Taq DNA polymerase (NEB, Ipswich, MA). This PCR step allowed for the amplification of the ligated DNA fragments with the addition of barcode sequences. Subsequently, a second round of PCR amplification was performed using 6-carboxy fluorescein (6-FAM) tagged barcode DNA-specific primer and primer 3 (Supplementary Data 4). This step allowed for the incorporation of 6-FAM-tag into the PCR products for downstream analysis. The 6-FAM-tagged PCR products were then separated by capillary electrophoresis at the FIU Sequencing Core, and their sizes were determined with DNA fragment analysis using GeneMapper V.5. Finally, the specific locations of SSBs at nucleotides in B-DNA and H-DNA-forming sequences in were mapped. The peak heights of the DNA strand breaks representing their abundance at specific nucleotides were added to calculate the DNA damage index (DDI), to represent the total SSB damage across the B-DNA and H-DNA-forming sequences in brain tissues of B-DNA and H-DNA mice on different diets.

### Ligation-mediated PCR assay

DSB locations at or near the H-DNA-forming sequences were mapped using LM-PCR on genomic DNA extracted from brain, liver, and testes tissues of both B-DNA and H-DNA mice on different diets as described previously[53]. Briefly, a 10 μM linker DNA was prepared by annealing LM-PCR1 and LM-PCR2 (Supplementary Data 4) in a 1:1 molar ratio in annealing buffer (10 mM Tris, 100 mM NaCl), followed by heating at 95 °C for 5 min and gradually cooling to room temperature. To convert DSBs with overhangs and recessed ends to blunt ends, 10 μg of genomic DNA was incubated with 10 U of DNA polymerase I large Klenow fragment (NEB, Ipswich, MA) and 33 μM dNTPs in a 40 μL reaction volume for 15 min at 25 °C. The reaction was then stopped by adding 10 mM EDTA and heat-inactivated for 20 min at 25 °C, followed by phenol-chloroform extraction and ethanol precipitation overnight at −20 °C.

The Klenow-treated DNA pellets were resuspended in 15 µL of 10 mM Tris-HCl pH 8.0 and ligated with 5 µL of 10 µM linker DNA using 1x blunt TA ligase (NEB, Ipswich, MA) for 45 min at 25 °C. The ligated DNA was then purified by phenol-chloroform extraction and overnight ethanol precipitation at −20 °C. Approximately 20% of the linker-ligated DNA resuspended in 10 µL of 10 mM Tris-HCl pH 8.0 was PCR amplified with 375 nM LMPCR2 linker-specific primer and 250 nM Lacleft 741 p2RT-specific primer (Supplementary Data 4) in 1x GoTaq master mix (Promega, Madison, WI). As positive controls, LM-PCR was performed on 20 ng of Sal1 (NEB, Ipswich, MA) digested B-DNA and H-DNA reporter DNA mixed with 100 µg of FVB mouse genomic DNA to mimic the approximate copy number of B-DNA or H-DNA, respectively, in the mouse genome. Negative controls included LM-PCR on FVB mouse genomic DNA, B-DNA-containing reporter DNA, H-DNA-containing reporter DNA, and PCR reaction mix without DNA.

The PCR-amplified products were resolved on 1.2% agarose gels and stained with SYBR gold. The DNA bands of interest were excised from the gels and recovered using the QIAquick gel extraction kit (Qiagen, Germantown, MD). The recovered DNA fragments were cloned using the CloneJet PCR cloning kit (Thermo Fisher Scientific, Waltham, MA) and plated on LB agar plates containing ampicillin (100 µg/L). Single white colonies were cultured in 5 mL LB media containing ampicillin (100 µg/mL) for 16 h at 37 °C and 2 × $g$ (250 rpm) in the orbital incubator shaker. The reporter DNA was retrieved using a QIAprep spin miniprep kit (Qiagen, Germantown, MD). Finally, the clones were submitted for Sanger sequencing, and the sequences were analyzed by comparing them against the reference cloning vector sequence using NCBI BLAST.

## Preparation of mouse tissue extracts for the DSB end-joining assay

Tissue extracts for DSB repair assays were prepared from frozen testicles from mice on both the CD and HFD diets following a modified version of the previously described method[97]. Briefly, a frozen testicle was washed in cold PBS, thawed, and resuspended in 300 µL of ice-cold lysis buffer (45 mM HEPES-KOH pH 7.8, 0.4 M KCl, 1 mM EDTA, 0.1 mM DTT, 10% glycerol) containing 1 mM phenylmethylsulfonyl fluoride (PMSF), 100 µM sodium orthovanadate (pH 10.5), and 1x protease inhibitor cocktail (Cell Signaling, Danvers, MA). The tissue was gently crushed using 50 strokes in a Dounce homogenizer. After 10 min of incubation on ice, the tissue extract was gently mixed with 30 µL of 1% Triton X-100 per 100 µL of the lysis buffer used. The tissue extract was left undisturbed on ice for 30 min. The extract was centrifuged for 15 min at 4 °C, 16,000 × $g$ until the supernatant was clear of any cell debris. The protein concentration (mg/mL) of the tissue extract was measured using the Lowry-based detergent-compatible protein assay (Bio-Rad, Hercules, CA). Finally, the extracts were aliquoted and stored at −80 °C until further use.

## DSB end-joining repair assay

The efficiency of testes tissue extracts from mice on the CD and HFD diets to repair DSBs was performed using a DSB end-joining repair assay developed by Kompella, del Mundo, & Vasquez (unpublished). Integrated DNA Technologies, Inc. (IDT), Coralville, IA, USA (Supplementary Data 4) synthesized the necessary oligonucleotides used for the assay. Briefly, a fluorophore-labeled oligonucleotide (250 nM) was annealed with unlabeled oligonucleotides to simulate DSBs with either 5′-5′ compatible ends (5′CE) or 5′-5′ non-compatible ends (5′NCE). The duplexes were then incubated with 150 µg of testicular tissue extract for 4 h at 37 °C. Each 25 µL reaction contained 1x NHEJ buffer (45 mM HEPES−KOH pH 7.9, 7.5 mM MgCl₂, 1 mM DTT, 2 mM ATP, 50 µM dNTPs, 10 µg/mL BSA, 2% glycerol) supplemented with an ATP regenerating system (40 mM phosphocreatine and 1.5 U freshly prepared creatine phosphokinase in 1% BSA). Reactions were stopped by incubating them with 10 mM EDTA and 40 µg of RNase A for 10 min at 37 °C and subsequently with 1% SDS and 25 µg of proteinase K for 15 min at 50 °C. The reaction products were purified by phenol/chloroform extraction and ethanol precipitation in the presence of 20 µg glycogen overnight at −20 °C.

On the following day, the repair products were resuspended in 8-10 µL of formamide denaturing buffer (10 mM EDTA, 1.5% Ficoll, 8% nuclease-free water, and 80% formamide), heated at 95 °C for 5 min and immediately put on ice. Without loading dye, the repair products were resolved on an 8% urea-denaturing polyacrylamide gel electrophoresis (PAGE). Following previous protocols, a 50-bp DNA ladder (NEB, Ipswich, MA) labeled with Cy-5 dCTP (Amersham, Cytiva, Marlborough, MA) was included as a size reference[98]. After electrophoresis for 55 min at 50 °C, 55 W, the gel was subsequently transferred onto a glass plate of the ChemiDoc MP (Bio-Rad, Hercules, CA) biomolecular imager for visualization using a red laser at a wavelength of 635 nm.

## Measurement of DSB end-joining repair efficiency in mouse tissue extracts

The end-joining repair efficiency was determined by calculating the integrated density of the shorter-length (100–150 bp, blue box) and longer-length (450-500 bp, red box) repair products relative to the full-length region. The captured images were analyzed using the National Institutes of Health (NIH) ImageJ 1.8.0_172 software. For each of the biological replicates from B-DNA mice on the CD and the HFD, the end-joining repair efficiency was quantified by measuring relative integrated density at two regions of interest on the gel. A consistent window width of 36 pixels was used for all samples, while the window heights varied based on the specific regions of interest. Three integrated density measurements were obtained for each sample, and their averages were recorded. To determine the end-joining efficiency for DNA substrates with compatible and non-compatible ends, three regions of interest were selected: the full-length region, the region corresponding to the shorter repair products (100–150 bp), and the region corresponding to the longer repair products (450–500 bp). Subsequently, the relative integrated intensity was calculated by dividing the integrated densities for a specific region by the integrated intensity of the full-length region. This calculation provided a measure of the relative contribution of each repair product to the entire repair process. To assess the impact of the HFD on end-joining repair efficiency, the percent increase or decrease was calculated relative to the CD group.

## Western blot analysis

Mouse testicular tissue extract (50 µg) was separated by SDS-PAGE on 4-15% Criterion™ TGX™ gel (Bio-Rad Hercules, CA) at 120 V for 80-90 min. The proteins were transferred to a nitrocellulose membrane using the Trans-Blot® Turbo™ Transfer System (Bio-Rad, Hercules, CA) for 7-15 min. The blots with the respective target proteins were blocked in 3.5% blotting grade dry milk (Bio-Rad, Hercules, CA) in 1x Tris-buffered saline (TBST; 0.02 M Tris base, 0.15 M NaCl, and 0.05% Tween 20) except for Mre11 and Rad 50 which were blocked in 3% BSA in 1x TBST for an hour at room temperature. The blots were probed overnight at 4 °C with the following antibodies diluted in 3.5% milk in 1x TBST: Ku70 (1:4000, A7330, Abclonal, Woburn, MA), DNA-PK (1:500, MA5-13238, Invitrogen, Carlsbad, CA), XRCC4 (1:500, GTX109632, Genetex, Irvine CA), Ligase IV (1:500, NBP2-16182, Novus Biologicals, Littleton, CO), XRCC1 (1:500, A4135, Abclonal, Woburn, MA), Ligase III (1:500, NBP1-41190, Novus Biologicals, Littleton, CO); β-actin (1:2000, QAB10339, EnQuirebio), γ-H2AX (1:5000, A300-081, Bethyl Laboratories, Montgomery, TX), and Vinculin (1:2000,13901, Cell Signaling, Danvers, MA). Antibodies against Mre11 (1:1000, NB100-142, Novus Biologicals, Littleton, CO), Rad 50 (1:500, GTX32832, Genetex, Irvine, CA), and OGG1 (1:500, NB100-106, Novus Biologicals, Littleton, CO) were diluted in 3% BSA in 1x TBST. Subsequently, the blots were incubated with the corresponding unconjugated host secondary

antibody for 2 h at room temperature in the dark. After incubation with ECL substrate (Bio-Rad, Hercules, CA) for 10–30 min in the dark, the chemiluminescence signal was detected using ChemiDoc Imaging System (Bio-Rad, Hercules, CA). Densitometry analysis was performed using NIH ImageJ 1.8.0_172 software with the band/peak quantification macro tool[99]. The level of a protein of interest was expressed as its signal intensity relative to a housekeeping gene, Vinculin or β-actin divided by 1000. The normalized values represent the average of 4–5 biological replicates (mean ± SEM).

## Statistical analyses

The data are presented as mean ± SEM (standard error of the mean) for biological replicates. Before statistical analysis, the normality of the quantitative datasets was assessed. Statistical analyses were performed using GraphPad Prism v9 (GraphPad Software, LLC). For two-group comparisons, an unpaired, two-tailed Student's $t$ test was utilized with or without Welch's correction, depending on the dataset. The results of the $t$-test are reported as t-statistic, degrees of freedom and p-value. Multiple-group comparisons were conducted using two-way ANOVA, followed by Sidak's multiple comparisons post hoc tests. The interaction between diet and B-DNA- or H-DNA-forming sequence in the mouse chromosomes and their simple main effects were assessed and reported as F-statistic and p-value. Western blot analysis was performed using the Mann–Whitney $U$-test. A significance threshold of $p$-value < 0.05 was employed to determine statistical significance.

## Reporting summary

Further information on research design is available in the Nature Portfolio Reporting Summary linked to this article.

## Data availability

All data supporting the findings of this study are available within the article and its supplementary information files and supplementary data files. Source data are provided with this paper and other relevant information are available in Figshare with the DOI identifier: https://doi.org/10.6084/m9.figshare.23820834.

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

## Acknowledgements

This study was supported by an NIH/NCI grant to K.M.V. (CA093729), NIH/NCI grant to K.M.V. and J.D. (CA225029), CPRIT Core Facility Support Award (RP170002) to K.M.V., the American Foundation for Pharmaceutical Education Pre-Doctoral Fellowship and UT Austin Graduate School Dissertation Writing Fellowships to P.K. We would like to thank Dr(s). John Powers, Imee del Mundo, Pooja Mandke, Anirban Mukherjee, Rick Finch, Dawit Kidane, and late Edward Mills for their valuable suggestions on this study.

## Author contributions

Conceptualization by K.M.V., P.K., G.W., and J.D; Methodology by P.K., G.W., Y.Lai, Y.Liu, and K.M.V.; Investigation by P.K., Y.Lai and C.M.; Formal analysis by P.K., R.E.D., C.M., Y.Liu, and S.L.H; Writing by P.K. and K.M.V.; Editing by P.K., K.M.V, G.W., and J.D., Supervision by G.W. and K.M.V; Funding acquisition by K.M.V., J.D., and P.K.

## Competing interests

The authors declare no competing interests.
