## [Peer Review File · Nature Communications]

Obesity increases genomic instability at DNA repeat-mediated endogenous mutation hotspotsREVIEWER COMMENTS

Reviewer #1 (Remarks to the Author):

Here, Kompella and colleagues begin to address an important problem: obesity is associated with increased cancer risk, but it is not at all clear why. In all likelihood, the reasons for this association are multifactorial, but one plausible component factor could be the obese tissue environment impacts upon genome stability. Therefore, there is no doubt this is a question that merits investigation. The only realistic way of proceeding to perform such studies in a well-controlled manner is in animal models, here the mouse is employed. Their approach was to create transgenic mice harbouring a reporter allele that either contains a (presumed) native 'B' type DNA structure and compare this in an isogenic relative that contains short purine-rich mirror repeats. Using a modification of the long-established LacZ/SupF system that allows blue white mutation screening and subsequent sequence analysis.

The first 3 figures in the paper that deal with the genetic aspects are of substantial interest, whereas the attempts to decipher the mechanics basis of the reported phenomena are rather less convincing.

Main points:

1. In the Introduction, there is a sense that the especially in paragraph 2 and 3 of page 3, that all the information described forms a narrative around obesity and its role genome instability. It probably wasn't the authors' intention, but it is important to be clear that this is not the case: for example, to someone new to the field, it might appear that refs 23 and 24 are related and that bile acid induced damage has been shown to produce translations, which is not what is reported there (23 is a review about bile acid and cellular stress, and 24 is a review on DSB dynamics). If that link has been made then please cite the primary references. The second paragraph of page 3 has similar issues.
2. I am aware that the corresponding author's group has a long history of studying triplex/H-DNA type structures and this is an interesting field. Notwithstanding, the mirror repeat sequences reported could also other transient intramolecular structures that could be lead to genomic rearrangements. I feel this should be discussed and the paper be less dogmatic on the point that the observed sequence instability must be due to H-DNA formation.
3. While I find the data presented in Figs 2 and 3 to be reasonable, Fig. 4 onwards a more problematic. In Fig 4A and B, a modest increase in 8-OxoG in HFD reported/quantified in 4A using an immunoassay, but the signal is almost gone on 4B what is the explanation?
4. There is an overall sense that the conclusion of this section is that H-DNA is more susceptible to 8-OxoG formation and this might underlie the fragility of these sequences. This is untested in the manuscript and is not a conclusion that can be substantiated. It could be tested, at least in vitro, relatively easily.
5. Page 15/Fig. 4C. The reference (66) is thin on detail and some more is provided in Methods for the current paper. It is unclear how well-validated this is – do we really know what is detected? Moreover, why does the HD lead to a reduction in damage at the B-DNA locus? The authors that there is no difference, but there certainly appears to be one.
6. Fig. 4 jumps around from tissue-to-tissue (liver-to-brain in 4A and 4B, then several tissues in 4E-H to liver). Why is this? Is it that some of the assays only showed differences in specific tissues? If so, the authors need to be clear on this point.
7. Fig. 4E-H: I am surprised at the level of breakage reported, even taking into account amplification. I am also surprised by the qualitative differences across tissues. This point would really need to be

followed up by orthogonal approaches: γ H2AX/53BP1 staining of tissues coupled to allele-specific probes, or similar.

8. Fig. 4I/J. Immunoblots are semi-quantitative. They are useful for detecting the presence and absence of proteins and large changes in quantity or for protein modification. Given all the issues with normalization these small changes have to be treated with great caution. Have the authors performed Q-RT-PCR on these factors? The regulation might not be at the transcriptional level, I accept, but at least it is quantitative. This is a problematic aspect of the reported work.

9. The end-joining assay shown in Figure 5 appears to give very complex collection of reaction products from the gels shown. I would have substantial reservations about drawing conclusions from this. These types of gel, where many species are generated are incredibly difficult to reliably quantify and normalise, again, the small differences reported. For example, a 12.8% reduction as reported in Fig. 5E might have (slight) statistical significance it doesn't mean there is any compelling biological difference given the qualitative complexities of the assay and its analysis, and that the sample size is not huge. The move to testicular is partly justified, although such assays do work in other tissues.

10. Fig. 6: again, I have substantial reservations of calling the significance of – at best – 2-fold changes using a semi-quantitative assay. Moreover, what could this mean mechanistically? There are nearly 0.5 million of copies of KU70 and KU80 in mammalian cells, so would a two-fold increase have any impact given the tiny number of endogenous DSB encountered in most somatic tissues at any one time? Almost certainly not. Moreover, the proteins studied here are not generally regulated within the pathways they operate at the expression level, but via interactions and post-translational modification. For any immunoblot quantification it is probably not necessary/appropriate to use a violin plot. It would be preferable to see some kind of plot, violin or otherwise, where the data points are also shown.

Minor points:

1. The Introduction section is long and contains quite a bit of methodological detail that should be in the Results section (page 4).
2. The figure panels are pretty tiny, especially when printed out. The panels Fig. 4, for example, would be too small even for a final published paper.
3. For those with limited knowledge of the experimental nutrition of mice, please define what is meant by kcal% fat (page 6) It seems like a unit that needs some clarification.
4. Define 'DIO' on first usage (page 6).
5. Page 14, the figure panels in Fig. 4 are not dealt with in order: we jump from 4A/B right to 4I/J.

Reviewer #2 (Remarks to the Author):

In this manuscript, the authors investigate the role of obesity in DNA genomic instability. The conclusions of the study are based on transgenic male mice exposed to diet-induced obesity. The mouse model carried a human H-DNA-forming or B-DNA (control) insert sequence from a translocation breakpoint hotspot in the c-MYC gene in Burkitt lymphoma. The authors focused on the brain, liver, testis to address the effects of obesity in genomic instability.

The authors demonstrated that mutation frequencies in genomic DNA in brain, liver, and testis were significantly affected by high-fat feeding in both B-DNA and H-DNA-mice but have a more pronounced effect in H-DNA mice. The authors showed that the brain had the highest accumulation of both point

mutations and large deletions.

In addition to the lacZ-based mutation assays, the authors studied genomic DNA from mouse liver tissue to confirm their findings. It was observed that obesity significantly increased the oxidative DNA damage marker levels, 8-oxo-dG, in steatotic livers. The results suggested that the unique structural vulnerabilities of H-DNA may result in increased oxidative DNA damage, leading to single-strand breaks in obesity. Finally, they examined DNA repair mechanisms, and found altered expression of proteins involved in non-homologous end-joining repair. Overall, this is a comprehensive study that contributes to the already established notion of DNA damage and cancer predisposition induced by obesity.

Major points to be addressed by the authors:

- 1) The study's relevance will be improved if genomic instability is demonstrated in the different cell types. For example, will obesity differentially affect immune cells or hepatocytes in the development of liver dysfunction? Hepatocytes can be easily isolated; the authors should address whether DNA changes are observed in these cells contributing to hepatocellular carcinoma.
- 2) The authors focused on male mice. The use of females is absolutely relevant to dissect any possible gender effect of the diet in genomic instability.
- 3) Are mutation frequencies in genomic DNA affected by short-term exposure to obesity? Moreover, based on the hypothesis of the authors, the DNA damage should be reversible in the liver, which is a tissue that has a high cell turnover rate. The authors should perform an experiment of changing to a low-fat diet to confirm this assumption.
- 4) Figure 6. The Western blots have been unclearly cropped (from the original files). The comparable samples should be loaded in the same blot to allow protein quantification.
- 5) How can this study be integrated in different human cancer risks associated to obesity?

Minor points

- 1) Figure 1 represents the research strategy and should be described in the methods section and moved to the Supplementary information.
- 2) Figure 2, please include stars in the panels.

Reviewer #3 (Remarks to the Author):

The manuscript, "Obesity increases genomic instability at DNA repeat-mediated endogenous mutation hotspots" by Kompella et al presents finding on the differential susceptibility of integrated H-DNA vs B-DNA sequences to point and deletion mutagenesis in mice on a high-fat vs standard chow diet. The investigation also probes underlying mechanisms that drive the increased mutation frequencies and the influences of end-joining processes under various physiological conditions. Although the overall conclusions seem largely supported by the data presented, in the following, there are several aspects of the investigation that need additional work and clarification.

In the Results section, additional data characterizing the founder and offspring B-DNA and H-DNA mice needs to be added. Given the procedures through which these mice were created, it would be anticipated that there could be significant differences in both copy number of integrations and the sites of integration. Authors should supply a more robust characterization of these strains. Further, it would be anticipated that there would be multiple founder mice for each genotype – were data obtained from multiple founders, and if so how comparable are the data within a genotype versus

comparisons made between genotypes?

Concerning the sequencing data for point mutations and deletions, of the 446 bp paired-end Illumina sequencing data that covered the B- and H-DNA regions, it would be useful to know the overall frequency within the H-DNA region. Also, the data in Fig 3 E (which appears to show point mutations even though the title to the figure legend indicates that this is for deletions) and S2 report bp positions 1-446, but the Tables 1-3 report frequency positions ranging from ~600 to ~1400 – it was not clear how to relate the data in the figures with the data in the tables.

For the section, "Obesity increases oxidative DNA damage in mice", the text flows from Panels A & B to Panels I & J – the text description flows well, but it seems that the figure should be reorganized to flow Panels A-D. Data presented in the current Fig 4 Panels C-H should be a separate figure.

The interpretation of data presented in Fig 5 D & E could significantly influenced by data concerning mouse #5 (B-DNA mice on HFD) – the majority of the input substrate appears to have remained intact. It is also challenging that the signal/noise ration in this assay is sufficient to report an 8.4% decrease in specific ligation products. It was also surprising that the overall efficiency of the ligation using DNAs with compatible ends did not appear to be significantly higher than reactions using DNAs with noncompatible ends (comparison of images in Fig 5 Panels D & G). The authors should provide insights into why these extracts do not efficiently catalyze compatible-end ligations. Overall, conclusions drawn from these data ("suggest that in the obsess state, DNA end-processing of such intermediates may be compromised") may be premature.

Concerning the data presented in Fig 6, it seems unlikely that the integration of one or more linear plasmids (B- vs H-DNA mice) into the entire mouse genome would make any difference in the overall expression levels of proteins involved in DSB end processing/joining proteins – however, the differences that might be expected to be created under conditions of CD vs HFD would seem reasonable to test. Thus, even though the authors report some individual protein expression differences between B- vs H-DNA mice, limiting the analyses to such a presentation of the low vs high fat diet data seems more hypothesis driven and justifiable.

Minor

The flow of information within the Introduction seems a bit disjointed – the text given between lines 90-114 seem to be at a basic general background level, and could be placed immediately after the opening paragraph, after line 44.

In discussion of the characterization of mouse H-DNA sequences, the paper by Maekawa, K et al PNAS 119, 19 on mapping H-DNA sequences in the mouse genome needs to be added.

Response to Referees

Reviewer #1 (Remarks to the Author):

“Here, Kompella and colleagues begin to address an important problem: obesity is associated with increased cancer risk, but it is not at all clear why. In all likelihood, the reasons for this association are multifactorial, but one plausible component factor could be the obese tissue environment impacts upon genome stability. Therefore, there is no doubt this is a question that merits investigation. The only realistic way of proceeding to perform such studies in a well-controlled manner is in animal models, here the mouse is employed. Their approach was to create transgenic mice harbouring a reporter allele that either contains a (presumed) native ‘B’ type DNA structure and compare this in an isogenic relative that contains short purine-rich mirror repeats. Using a modification of the long-established LacZ/SupF system that allows blue-white mutation screening and subsequent sequence analysis.

The first 3 figures in the paper that deal with the genetic aspects are of substantial interest, whereas the attempts to decipher the mechanics basis of the reported phenomena are rather less convincing”.

Main points:

1. **“In the Introduction, there is a sense especially in paragraphs 2 and 3 of page 3, that all the information described forms a narrative around obesity and its role in genome instability. It probably wasn’t the authors’ intention, but it is important to be clear that this is not the case: for example, to someone new to the field, it might appear that refs 23 and 24 are related and that bile acid induced damage has been shown to produce translations, which is not what is reported there (23 is a review about bile acid and cellular stress, and 24 is a review on DSB dynamics). If that link has been made then please cite the primary references. The second paragraph of page 3 has similar issues.”**

Response: We value the insightful feedback provided by the reviewer. We have revised paragraphs 2 and 3 (in the original manuscript) to improve the clarity and accuracy of our introduction section. Additionally, we have replaced review articles with research articles wherever applicable to avoid any potential confusion for readers new to the field.

For clarity, the introduction and references were modified as follows:

Obesity-associated cancers involve disruptions in multiple metabolic and cellular pathways²⁷. Some of these pathways converge on genomic instability, thereby increasing cancer risk^{28, 29}. Moreover, alterations in metabolism, adipose tissue dysfunction, hormonal imbalances, and chronic inflammation, can lead to systemic oxidative stress^{30, 31}. Apart from disrupting cellular and tissue homeostasis³⁰, oxidative stress can reduce antioxidant defense mechanisms and increase the generation of reactive oxygen species (ROS), consequently increasing the levels of mutagenic oxidized DNA lesions, such as 8-oxo-7,8-dihydro-2'-deoxyguanosine (8-oxo-dG), among others^{32, 33}. Recent research indicates that single-strand DNA breaks (SSBs) associated with unrepaired 8-oxo-dG lesions can lead to double-strand breaks (DSBs) in DNA, contributing to genomic instability³⁴.

In addition, obesity-inducing dietary fat can stimulate gut dysbiosis³⁵, altering the metabolism of liver-derived primary bile acid metabolites into secondary bile acid metabolites³⁶. The formation of these

secondary bile acid metabolites is associated with oxidative stress resulting in DSBs^{37, 38}. Persistent DSBs, may cause deletions and chromosomal abnormalities, including cancer-associated translocations, such as those observed in the human *c-MYC* gene implicated in Burkitt lymphoma, and other cancers^{39, 40, 41}.

2. **“I am aware that the corresponding author’s group has a long history of studying triplex/H-DNA type structures and this is an interesting field. Notwithstanding, the mirror repeat sequences reported could also other transient intramolecular structures that could be lead to genomic rearrangements. I feel this should be discussed and the paper be less dogmatic on the point that the observed sequence instability must be due to H-DNA formation.”**

Response: The reviewer makes a good point, and we have revised the text accordingly. For example, we refer to the sequence as an “H-DNA-forming” sequence rather than referring directly to the structure “H-DNA”. We feel that referring to it as an “H-DNA-forming” sequence is justified based on our many years of studying this (and other) potential H-DNA-forming sequences. This sequence in particular has been extensively studied by our group to address the same question that the reviewer has asked, and through much characterization using a variety of computational, biophysical, chemical, and molecular techniques *in silico*, *in vitro*, and *in vivo*, we are quite confident that this particular sequence does indeed form H-DNA, at least *in vitro*. For example, please see Belotserkovskii et al, *JBC*, 2007 and Del Mundo et al, *NAR*, 2017. Further, the control sequence contains changes to only 2 base pairs in this sequence, maintaining the G-richness (e.g. potential G4 DNA formation), while disrupting the mirror symmetry required for H-DNA formation, and based on a variety of techniques (as mentioned above), this mutated sequence does not form non-B DNA structures of any kind, but instead forms canonical B-DNA, as predicted.

3. **“While I find the data presented in Figs 2 and 3 to be reasonable, Fig. 4 onwards a more problematic. In Fig 4A and B, a modest increase in 8-OxoG in HFD reported/quantified in 4A using an immunoassay, but the signal is almost gone on 4B what is the explanation?”**

Response: We appreciate the valuable feedback from the reviewer and acknowledge the concern raised regarding the 8-oxoG signal intensity between Fig. 4a and 4b. Upon careful examination, we identified that the issue stemmed from the contrast and brightness of the images. In response to this, we have rectified the matter by uniformly adjusting the brightness and contrast of all the images, ensuring a consistent and fair comparison across all panels within Fig. 4b. These changes are considered linear (i.e., all pixels become brighter) as it was applied to the entire image and these adjustments have been made to more accurately represent the data, and we trust that the revised figures now provide a clearer and more reliable presentation of the 8-oxoG signal.

For clarity, this change was mentioned in the legend of Fig. 4b:

The images were contrast-enhanced for clarity.

4. **“There is an overall sense that the conclusion of this section is that H-DNA is more susceptible to 8-OxoG formation, and this might underlie the fragility of these sequences.**

This is untested in the manuscript and is not a conclusion that can be substantiated. It could be tested, at least in vitro, relatively easily.”

Response: This is an insightful observation by the reviewer. Subsequently, our laboratory has undertaken an *in vitro* investigation to test this hypothesis. Reporter plasmids containing either a B-DNA or an H-DNA-forming sequence were exposed to oxidative stress *in vitro*, and slot blot analysis was performed on H-DNA-containing and B-DNA-containing restriction fragments. The results demonstrated a significant increase (~2-fold) in 8-oxoG lesions in the H-DNA fragment compared to the B-DNA fragment. The results from this study have been submitted as a separate research article currently under review. These findings provide support for the conclusion that H-DNA is more susceptible than B-DNA to 8-oxoG lesion formation following oxidative stress, potentially contributing to the observed fragility of these sequences.

To enhance clarity, the following sentences were incorporated into the Discussion section:

In a recent study, we subjected B-DNA and H-DNA reporter plasmids to oxidative stress in mammalian cells *in vitro*. Our findings (unpublished) revealed that H-DNA-forming sequences demonstrated increased susceptibility to 8-oxo-dG formation compared to B-DNA-forming sequences offering a potential explanation for the inherent mutability of these sequences.

5. **“Page 15/Fig. 4C. The reference (66) is thin on detail, and some more is provided in Methods for the current paper. It is unclear how well-validated this is – do we really know what is detected? Moreover, why does the HD lead to a reduction in damage at the B-DNA locus? The authors that there is no difference, but there certainly appears to be one.”**

Response: We thank the reviewer for this comment. The assay is currently under review for patent application by our collaborator and co-author, Dr. Yuan Liu at Florida International University. This prevented us from providing a very detailed description of the DNA modification landscape assay. Our technique specifically detected ssDNA breaks (SSBs) generated endogenously from oxidative DNA damage 8-oxoG, AP sites, nicks, and gaps in B-DNA- and H-DNA-forming sequences from brain tissues of B-DNA and H-DNA mice on different diets. The locations of SSBs on the strands of specific genes and genomic DNA regions were validated by purifying, cloning, and sequencing the PCR products amplified using a gene-specific primer and barcode primer. According to this reviewer’s comments, we have now provided more information about the assay by modifying the description in the methods section of the revised manuscript.

Moreover, in our study, the statistical analysis showed that there was no significant difference in DDI in brain tissue between B-DNA mice on CD and HFD ($p>0.05$) (Refer to the results section: Obesity exacerbates the formation of H-DNA-induced SSBs and DSBs in mice). However, it should be noted that mouse brain tissue from the B-DNA mice on the HFD exhibited a lower DDI than those on the CD (Fig. 5A). These results may suggest that the genomic region containing the B-DNA-forming sequence was protected from endogenous oxidative DNA damage in mouse brain tissue in response to oxidative DNA damage induced by HFD. This may be due to chromatin structural changes that shielded the B-DNA-forming sequence in the mouse genome, preventing DNA damage from HFD-induced oxidative stress. Interestingly, our results showed that the accumulation of SSBs and mutations were significantly increased in the brain tissue of H-DNA mice on the HFD (Fig. 5A)

compared to those on the CD. We have revised the discussion section to better describe potential explanations of the results obtained.

The description in the Methods section was revised as shown below:

DNA modification landscape assay

Our collaborator, Dr. Yuan Liu at Florida International University (FIU) developed the DNA modification landscape assay⁵⁹ used in this study to detect SSBs and DSBs in B-DNA- and H-DNA forming sequences in brain tissue of the B-DNA and H-DNA mice on different diets. Using this assay, SSBs resulting from endogenous oxidative DNA damage 8-oxo-dG, AP sites, nicks, and gaps on the 23-bp H-DNA-forming sequence and a corresponding B-DNA-forming sequence in the brain tissue of B-DNA and H-DNA mice on the different diets were detected. Briefly, 2 µg of genomic DNA was denatured at 95°C for 5 minutes resulting in single-stranded DNA. Subsequently, primer extension was performed using a 23-nucleotide primer (primer 1) specifically annealed to the downstream region of the B-DNA and H-DNA forming sequences (Supplementary Table 4) using vent DNA polymerase (NEB, Ipswich, MA). This step generated double-stranded DNA fragments with various sizes that represent different sites of SSBs on the target B-DNA and H-DNA-forming sequences. Next, the DNA fragments were ligated with a barcode DNA and subjected to PCR amplification using primer 2 (annealed to the downstream region closer to B-DNA or H-DNA-forming sequences than primer 1) (Supplementary Table 4) and the barcode primer (IDT, Coralville, IA) (Supplementary Table 4) with the LongAmp Taq DNA polymerase (NEB, Ipswich, MA). This PCR step allowed for the amplification of the ligated DNA fragments with the addition of barcode sequences. Subsequently, a second round of PCR amplification was performed using 6-carboxy fluorescein (6-FAM) tagged barcode DNA-specific primer and primer 3 (Supplementary Table 4). This step allowed for the incorporation of 6-FAM-tag into the PCR products for downstream analysis. The 6-FAM-tagged PCR products were then separated by capillary electrophoresis at the FIU Sequencing Core, and their sizes were determined with DNA fragment analysis using GeneMapper V.5. Finally, the specific locations of SSBs at nucleotides in B-DNA and H-DNA-forming sequences in were mapped. The peak heights of the DNA strand breaks representing their abundance at specific nucleotides were added to calculate the DNA damage index (DDI), to represent the total SSB damage across the B-DNA and H-DNA-forming sequences in brain tissues of B-DNA and H-DNA mice on different diets.

6. **“Fig. 4 jumps around from tissue-to-tissue (liver-to-brain in 4A and 4B, then several tissues in 4E-H to liver). Why is this? Is it that some of the assays only showed differences in specific tissues? If so, the authors need to be clear on this point.”**

Response: We appreciate the thoughtful observation made by the reviewer regarding the organization of Fig. 4. Performing the blue-white mutagenesis assay and LM-PCR assay on tissues isolated from obese mice posed technical challenges due to their high lipid content. Consequently, brain and testes tissue were exclusively flash frozen, while the larger liver size allowed for a combination of flash freezing and formalin fixation for mutation screening, and immuno-staining. This is why we could not perform all the assays in all the tissues.

We have also restructured the figure to enhance clarity. Fig. 4 now comprises panels a (8-oxo-dG: immunoassay), b (8-oxo-dG: Immunohistochemistry), c (OGG1: Western blotting), and d (OGG1:

Western blotting quantification) to create a more cohesive flow. Additionally, previously Fig. 4 panels c-h, i (WB- γ -H2AX), and j (quantification of γ -H2AX) have been separated into a distinct Fig. 5. The original Fig. 5 has been renumbered as Fig. 6. The original Fig. 6 has been renumbered as Fig. 7. We believe this adjustment enhances the logical presentation of the data and improves overall figure readability.

The Results section was modified to reflect the figure number changes and clarification for using the liver tissue as follows:

Obesity increases oxidative DNA damage in mice. We sought to explore an additional factor, 8-oxo-dG lesions, that could contribute to obesity-mediated genomic instability, in addition to the presence of endogenous H-DNA-forming sequences. Performing the blue-white mutagenesis assay on tissues isolated from obese mice posed technical challenges due to their high lipid content. Consequently, brain and testes tissue were exclusively flash frozen, while the larger liver size allowed for a combination of flash freezing and formalin fixation for mutation screening, and immunostaining. To assess levels of 8-oxo-dG in genomic DNA, liver tissues from mice on either the CD or the HFD were subjected to immunoassay and immunostaining techniques. Analysis of the immunoassay results revealed a significant increase ($p=0.0029$) in the oxidative DNA damage marker levels, 8-oxo-dG, in mice on the HFD compared to those on the CD (Fig. 4a). Furthermore, immunostaining of the liver tissue corroborated these findings, as illustrated in Fig. 4b.

In humans, 8-oxo-dG lesions are typically repaired via the base-excision repair (BER) pathway. The primary enzyme responsible for the initial recognition and excision of the lesion is 8-oxo guanine glycosylase (OOG1)⁵⁸. In response to increased oxidative DNA damage, we observed a significant increase ($p=0.0079$) in the levels of OGG1 protein in B-DNA mice on the HFD compared to those on the CD (Fig. 4c, d). A similar difference was not observed in the H-DNA mice on HFD compared to those on CD, perhaps due to the significantly elevated ($p<0.05$) OGG1 level in H-DNA mice regardless of the diet.

7. **“Fig. 4E-H: I am surprised at the level of breakage reported, even taking into account amplification. I am also surprised by the qualitative differences across tissues. This point would really need to be followed up by orthogonal approaches: γ H2AX/53BP1 staining of tissues coupled to allele-specific probes, or similar.”**

Response: We appreciate this comment, and we too were surprised by the number of breaks determined by LM-PCR, though we have repeated the experiments with similar results and have included appropriate positive and negative controls for the assay. As mentioned in response to comment #6 above, performing the blue-white mutagenesis assay and LM-PCR assay on tissues isolated from obese mice posed technical challenges due to their high lipid content. Consequently, brain and testes tissue were exclusively flash frozen, while the larger liver size allowed for a combination of flash freezing and formalin fixation for mutation screening, and immuno-staining. Therefore, doing an orthogonal assessment such as staining the brain and testes tissue for γ H2AX/53BP1 markers was not possible. However, since we utilized testis tissue for the DNA end-joining repair assay, we attempted to quantify protein levels of the DSB marker γ H2AX in the testicular tissue of B-DNA and H-DNA mice and observed a significant increase ($p=0.0159$) in the

levels of γ H2AX protein in B-DNA mice on the HFD compared to those on the CD (Fig. 5g and 5h). In contrast, the difference between the CD and the HFD groups of H-DNA mice was not statistically significant, partially due to the increased γ H2AX levels in H-DNA mice on even a CD. Nevertheless, the results suggested that the H-DNA-forming sequences and the obesity-inducing HFD diet could increase DNA breakage *in vivo*.

The Results section was modified to reflect the figure number changes and the limitation of the LM-PCR assay as follows:

Obesity exacerbates the formation of H-DNA-induced SSBs and DSBs in mice.

To investigate the mechanisms behind the elevated mutation frequencies and diverse mutation spectra observed in the H-DNA mice, we utilized a DNA modification landscape assay⁵⁹ to measure SSBs within and surrounding the H-DNA region in the brain tissue. This assay generates a DNA damage index (DDI) profile across the region of interest, allowing for the sensitive detection and quantification of DNA damage. Our analysis of DDI across the target B-DNA or H-DNA region (Fig. 5a) revealed that while there was no difference in HFD-induced DDI in B-DNA mice ($p=0.0557$) compared to those on the CD, the difference was significant in H-DNA mice ($p=0.0007$) when compared to their counterparts on the CD. Interestingly, we did not find a significant increase ($p=0.7839$) in H-DNA-mediated DDI alone when compared to B-DNA mice on the CD. However, obesity significantly increased ($p<0.0001$) the DDI in H-DNA mice compared to B-DNA mice. Our results suggest that the unique structural vulnerabilities of H-DNA may result in increased oxidative DNA damage, leading to SSBs in the obese state.

Interestingly, Ng et al. (2020) reported that somatic and meiotic DSBs tend to localize near the *MYC* oncogene⁶⁰. In our study, we observed increased frequencies of deletions within the H-DNA-forming sequences from the *c-MYC* gene in DNA isolated from the liver, brain, and testes tissues of obese mice. To gain further insights into the identification of DSBs within and surrounding the H-DNA-forming sequence, we employed linker-mediated PCR (LM-PCR) on mouse genomic DNA (Fig. 5b). We identified two DSB hotspots (at 250 bp and 300 bp) near the H-DNA sequence in DNA extracted from the liver tissue of obese H-DNA mice (Fig. 5c). Similarly, we detected three distinct DSB hotspots in the brain tissue DNA (at approximately 250 bp, 375 bp, and 750 bp) (Fig. 5d) and the testes tissue DNA (at approximately 250 bp) (Fig. 5e) from H-DNA mice on the HFD. We acknowledge the sensitivity limitation of this assay, as the DSBs hotspots i.e., accumulation of DSBs within a particular region were exclusively detected in obese H-DNA mice and not in obese B-DNA mice. However, these results align with the increased frequency of large deletions observed in the obese H-DNA mice compared to B-DNA mice. Sequence analysis confirmed that the DSBs occurred within and around the H-DNA-forming sequence (Fig. 5f). These findings support our hypothesis that metabolic changes in the obese state enhance H-DNA-induced SSBs and DSBs, resulting in increased genomic instability at these endogenous mutation hotspots *in vivo*.

As a cellular response to the induction of DNA damage, we also measured protein levels of the DSB marker γ -H2AX⁶¹ in the testicular tissue of B-DNA and H-DNA mice. We observed a significant increase ($p=0.0159$) in the levels of γ -H2AX protein in B-DNA mice on the HFD compared to those on the CD (Fig. 5g, h). In contrast, the difference between the CD and the HFD groups of H-DNA mice was not statistically significant, partially due to the increased γ -H2AX levels in H-DNA mice on even a CD. Nevertheless, the results suggested that the H-DNA-forming sequences and the obesity-inducing HFD diet could increase DNA breakage *in vivo*.

8. **“Fig. 4I/J. Immunoblots are semi-quantitative. They are useful for detecting the presence and absence of proteins and large changes in quantity or for protein modification. Given all the issues with normalization these small changes have to be treated with great caution. Have the authors performed Q-RT-PCR on these factors? The regulation might not be at the transcriptional level, I accept, but at least it is quantitative. This is a problematic aspect of the reported work.”**

Response:

While we appreciate the reviewer's concern regarding the semi-quantitative nature of immunoblots, we would like to emphasize that we chose to measure protein levels since they are the ultimate effectors of biological processes. We acknowledge that RT-PCR would provide information on the mRNA expression of the genes; however, Western blots provide additional insights into translation efficiency, post-translational modifications, and protein stability, offering a more comprehensive view of the protein landscape. As noted in studies such as Vogel et al. (*Mol. Syst. Bio.* 2010) and Maier et al. (*Mol. Syst. Bio.* 2011), the correlation between changes in gene expression at the mRNA level and protein abundance can vary, with some proteins showing a more direct correlation than the others. Our co-author on this manuscript, Dr. John DiGiovanni (and others, e.g. Dr. Stephen Hursting at UNC Chapel Hill) have published extensively on how dietary energy balance modulates signaling pathways. For example, in the publications Moore T. et al (*Cancer Prev Res (Phila)* 2008), Moore T. et al (*Cancer Prev Res (Phila)* 2012), and Lashinger LM et al (*Cancer Prev Res* 2011) using western blotting, they have shown that the obesity protocol used induces consistent changes in the cell cycle regulatory proteins, proteins involved in the Akt/mTOR signaling in multiple tissues and proteins involved in the IGF-1 signaling pathway. Thus, we think that this is an established method to obtain initial data on potential obesity-induced protein changes related to repair pathways.

In our study, the focus on proteins aligns with their functional role as key players in DNA repair pathways under investigation. We understand the importance of caution in interpreting small changes and potential issues with normalization. To address this concern, we have taken several steps, including using appropriate loading controls, consistent sample preparation techniques, a widely published quantification tool, and statistical analyses to ensure robust and reliable results. Further work is necessary to determine whether these changes in protein levels coupled with changes in post-translational modification and protein-protein interactions contribute to the mutational changes seen with obesity. We hope this clarification addresses the reviewer's concerns regarding the quantitative aspects of our work.

9. **“The end-joining assay shown in Figure 5 appears to give very complex collection of reaction products from the gels shown. I would have substantial reservations about drawing conclusions from this. These types of gel, where many species are generated are incredibly difficult to reliably quantify and normalise, again, the small differences reported. For example, a 12.8% reduction as reported in Fig. 5E might have (slight) statistical significance it doesn't mean there is any compelling biological difference given the qualitative complexities of the assay and its analysis, and that the sample size is not huge. The move to testicular is partly justified, although such assays do work in other tissues.”**

Response: We appreciate the reviewer's diligence in scrutinizing our experimental approach. The assay utilized in our study was adapted from the extensively published radioactivity-based method developed by Sathees C Raghavan's group. We transformed this method into a fluorophore-based system for enhanced sensitivity. While we acknowledge the qualitative complexities of the assay and the challenges in quantification, we think that it provides valuable insight into the trends of the end-joining repair process in the obese state relative to normal weight mice. Of note, the majority of studies reporting DSB repair assays in the literature use cell line protein extracts. Due to lipid deposition in tissues of mice fed a high-fat diet, extracting active protein from the liver tissue is exceptionally challenging. We would also like to point out that the cell-free extracts from tissues have active nucleases and other DNA repair/processing proteins that can potentially contribute to degradation of the un-ligated substrates and ligated products. Albeit unavoidable, this is an indicator of active and working cell-free tissue extracts. In light of this, we opted to use testicular tissue for our assay which has active ligation activity with moderate nuclease activity. However, we acknowledge the reviewer's suggestion, and for our future studies, we plan to utilize MMEJ/NHEJ reporter mice harboring B-DNA- and H-DNA-forming sequences, which would enable a more comprehensive analysis of repair mechanisms in different tissues.

We have now modified the Results section as follows:

Obesity reduces the end-joining repair efficiency of DSBs in mice.

It is worth noting that the cell-free extracts from tissues contain active nucleases and other DNA repair/processing proteins, which may contribute to the degradation of both un-ligated substrates and ligated products. Although unavoidable, this is an indicator of the activity and functionality of cell-free tissue extracts. Therefore, we utilized testicular tissue extract for this assay, known for its high DSB end-joining efficiency and moderate nuclease activity compared to other tissues^{62, 63}. During assay development, we observed that the kinetics of compatible end-joining was remarkably faster within the initial hour of substrate incubation in the tissue extract. However, prolonged incubation (>4h) to promote the formation of longer products resulted in DNA loss. Conversely, for substrates with non-compatible ends requiring processing, the kinetics appeared slower, with maximum multimerization occurring by 4h and further incubation resulting in DNA loss. To ensure consistency, both compatible and incompatible end substrates were incubated for the same duration of time (4h), facilitating a fair comparison between the two conditions.

We have now modified the Discussion section as follows:

While beyond the scope of the present study, future experiments utilizing NHEJ or MMEJ reporter mice⁸⁵ engineered to harbor B-DNA- or H-DNA-forming sequences and subjected to dietary interventions could provide comprehensive insights into the dynamic relationship between obesity, age, genomic instability, and cancer risk.

10. **“Fig. 6: again, I have substantial reservations of calling the significance of – at best – 2-fold changes using a semi-quantitative assay. Moreover, what could this mean mechanistically? There are nearly 0.5 million of copies of KU70 and KU80 in mammalian cells, so would a two-fold increase have any impact given the tiny number of endogenous DSB encountered in most somatic tissues at any one time? Almost certainly not. Moreover, the proteins studies**

here are not generally regulated within the pathways they operate at the expression level, but via interactions and post-translational modification.

For any immunoblot quantification it is probably not necessary/appropriate to use a violin plot. It would be preferable to see some kind of plot, violin or otherwise, where the data points are also shown.”

Response: We appreciate the reviewer’s comment and did not expect to see differences in protein levels in the H-DNA vs. B-DNA mice. Instead, we expected to detect differences in protein levels as a function of obesity in the mice. Predicting the precise impact of the observed two-fold increase in Ku protein levels in H-DNA mice compared to the B-DNA mice, particularly in the obese state is challenging. Our study suggests that Ku proteins primarily recognize DSBs without further participation in the NHEJ pathway. In previous studies, (Jain A et al. *Biochemie*, 2004 and Wang G et al. *PNAS*, 2004) we have demonstrated that DSBs induced by H-DNA are repaired via a MMEJ pathway, resulting in large-scale deletions with microhomologies at the breakpoint junctions. Consistently, we observed similar large-scale deletions with microhomologies in this study, along with the upregulation of proteins involved in MMEJ repair. In future investigations, we plan to do systematic immunodepletion experiments targeting key proteins in both the NHEJ and MMEJ pathways to elucidate the mechanisms underlying compromised repair efficiency, particularly in the obese state. However, if the reviewers feel strongly about this data, then we could remove it.

We have now modified Fig. 7 as follows:

Regarding the visualization of immunoblot quantification data, we initially used violin plots to depict the distribution of numerical data. Following the reviewer’s suggestion, we have now included data points in the plots for enhanced clarity and transparency.

11. Minor points:

- a) “The Introduction section is long and contains quite a bit of methodological detail that should be in the Results section (page 4).”

Response: We agree with the reviewer and have edited the Introduction section to make it more concise and the methodological details have been moved to the Results section.

We have now modified the Introduction section as follows:

To better understand the obesity-cancer link, we examined the impact of diet-induced obesity (DIO) on genomic instability at an endogenous cancer-associated mutation hotspot.

We conducted a targeted and quantifiable investigation by employing a transgenic mutation-reporter mouse model containing either a control B-DNA insert or a human H-DNA-forming sequence from a translocation breakpoint hotspot in the *c-MYC* gene in Burkitt lymphoma, herein referred to as B-DNA and H-DNA mice, respectively⁵¹. The characterization of the founder mice published previously by Wang et al. (2008) showed that despite the variations in copy number, the mutation frequencies within (B-DNA or H-DNA) and between (B-DNA vs. H-DNA) the mouse type remains remarkably consistent⁵¹. In the current study, B-DNA and H-DNA mice were selected based on the highest integrated reporter copy numbers (120-150), which is essential

for successful pull-down blue-white mutation screening. Thus, this model represents a powerful tool for detecting DIO-related mutations across multiple tissues simultaneously⁵².

Here, we tested the impact of obesity on H-DNA-induced mutagenesis in the liver, brain, and testes of male mice. These tissues were selected based on their proliferative potential because we have identified both replication-dependent and replication-independent mechanisms of non-B DNA-induced genetic instability⁵³. Our findings reveal an accumulation of point mutations, large deletions, SSBs, and DSBs enriched at mutation hotspots in a tissue-specific manner. Furthermore, we demonstrate that DIO compromises the efficient repair of DSBs. These findings offer valuable insights into how obesity modulates genomic instability mediated by DNA repeats, highlighting potential implications for obesity-related cancer therapies.

Based on the reviewer's comments we have modified the Results section as follows:

Tissue-specific impact of obesity on genomic instability.

To determine the impact of obesity on genomic instability, we measured mutation frequencies and analyzed mutation spectra generated in the p2RT mutation reporter harboring a control B-DNA-forming sequence (Fig. 1a) or an H-DNA-forming sequence (Fig. 1b) chromosomally integrated into mice (Fig. 1c). We utilized the binding of magnetic beads to the LacI-LacZ fusion protein on the chromosomally integrated mutation reporter to separate it from the mouse genomic DNA. After elution and self-ligation, a wide variety of mutations in the H-DNA compared to the control B-DNA region of the mouse genome were detected via a sensitive and facile blue-white mutagenesis assay in methylation-resistant DH10 β cells⁵² (Fig. 1d). For each diet group (CD and HFD) and tissue type (brain, liver, and testes) we counted a minimum of 20,000 colonies per each B-DNA and H-DNA mouse tissue. To characterize the mutation types contributing to genomic instability, 40-50 randomly selected mutants from each group were sequenced and compared with the respective B-DNA- or H-DNA containing p2RT reporter sequence using nucleotide BLAST. Our findings in Fig. 2 revealed intriguing tissue-specific differences in mutation patterns, with the brain and the liver showing a higher propensity for genomic instability than the testes tissue.

- b) **“The figure panels are pretty tiny, especially when printed out. The panels Fig. 4, for example, would be too small even for a final published paper.”**

Response: We apologize for any inconvenience during the review process due to the small font size. In response to this feedback, we have adjusted the figure size to comply with the specifications of the journal. We believe that these modifications will enhance the clarity and legibility of the figures, particularly when printed in the published paper.

- c) **“For those with limited knowledge of the experimental nutrition of mice, please define what is mean by kcal% fat (page 6) It seems like a unit that needs some clarification.”**

Response: Thank you for bringing this to our attention.

We have now modified the Results section as follows:

High-fat diet induces obesity and liver steatosis.

Male B-DNA and H-DNA mice were fed either a control diet (CD, 10 kcal% fat, representing 10% of total calories from fat) or an obesity-inducing high-fat diet (HFD, 60 kcal% fat, comprising 60% of total calories from fat) at 5-6 weeks of age for 13 weeks.

We have now modified the Methods section as follows:

Kcal% from fat is defined as the % of the total calories in that diet coming from fat.

d) “Define ‘DIO’ on first usage (page 6).”

Response: Thank you for bringing this to our attention.

We have defined Diet-induced obesity (DIO) on the first usage in the introduction section:

To better understand the obesity-cancer link, we examined the impact of diet-induced obesity (DIO) on genomic instability at an endogenous cancer-associated mutation hotspot.

e) “Page 14, the figure panels in Fig. 4 are not dealt with in order: we jump from 4A/B right to 4I/J.”

Response: We appreciate this observation regarding the organization of Fig. 4. In response, we have restructured the figure to enhance clarity. Fig. 4 now comprises panels a (8-oxo-dG: immunoassay), b (8-oxo-dG: Immunohistochemistry), c (OGG1: Western blotting), and d (OGG1: Western blotting quantification) to create a more cohesive flow. Additionally, previously Fig. 4 panels c-h, i (WB- γ -H2AX), and j (quantification of γ -H2AX) have been separated into a distinct Fig. 5. The original Fig. 5 has been renumbered as Fig. 6. The original Fig. 6 has been renumbered as Fig. 7. We believe this adjustment enhances the logical presentation of the data and improves overall figure readability.

Reviewer #2 (Remarks to the Author):

“In this manuscript, the authors investigate the role of obesity in DNA genomic instability. The conclusions of the study are based on transgenic male mice exposed to diet-induced obesity. The mouse model carried a human H-DNA-forming or B-DNA (control) insert sequence from a translocation breakpoint hotspot in the c-MYC gene in Burkitt lymphoma. The authors focused on the brain, liver, testis to address the effects of obesity in genomic instability.

The authors demonstrated that mutation frequencies in genomic DNA in brain, liver, and testis were significantly affected by high-fat feeding in both B-DNA and H-DNA-mice but have a more pronounced effect in H-DNA mice. The authors showed that the brain had the highest accumulation of both point mutations and large deletions.

In addition to the lacZ-based mutation assays, the authors studied genomic DNA from mouse liver tissue to confirm their findings. It was observed that obesity significantly increased the oxidative DNA damage marker levels, 8-oxo-dG, in steatotic livers. The results suggested that the unique structural vulnerabilities of H-DNA may result in increased oxidative DNA damage,

leading to single-strand breaks in obesity. Finally, they examined DNA repair mechanisms, and found altered expression of proteins involved in non-homologous end-joining repair. Overall, this is a comprehensive study that contributes to the already established notion of DNA damage and cancer predisposition induced by obesity”.

Major points to be addressed by the authors:

- 1) **“The study’s relevance will be improved if genomic instability is demonstrated in the different cell types. For example, will obesity differentially affect immune cells or hepatocytes in the development of liver dysfunction? Hepatocytes can be easily isolated; the authors should address whether DNA changes are observed in these cells contributing to hepatocellular carcinoma.”**

Response: This is an excellent suggestion from the reviewer, and though this is outside the scope of this study, we have proposed such studies in a new grant application and hope to secure funding to support those studies in the near future. The known association between persistent inflammation, lipid infiltration into the liver, and cellular damage in hepatocytes suggests a complex interplay that can induce oxidative stress, DNA damage, and subsequent compensatory repair processes, ultimately contributing to gene mutations and oncogenesis (Ioannou GN. *J. Hepatol.* 2021). Additionally, Yang J et al. (*Front. Immunol.* 2023) have recently provided a comprehensive review outlining distinct mechanisms by which obesity influences HCC progression, including the remodeling of the tumor immune microenvironment and induction of immunosuppression.

While the primary focus of our study was to investigate the impact of obesity on DNA repeat-mediated genomic instability, we acknowledge the relevance of exploring the specific effects on hepatocytes and immune cells in the liver. However, due to the constraints of using small-sized mouse tissues for investigating mutation burden in the current study, viable hepatocyte isolation from frozen liver tissue was not feasible. In future studies, we plan to conduct cell-type-specific analyses in the liver to provide a more in-depth assessment of how obesity may differentially affect genomic stability in various cell types and contribute to the development of liver dysfunction, including hepatocellular carcinoma.

The following sentence was included in the Discussion section:

In future studies, we aim to explore how obesity may differentially affect DNA repeat-mediated genomic stability in hepatocytes and immune cells within the liver, potentially contributing to the development of liver dysfunction, including hepatocellular carcinoma.

- 2) **“The authors focused on male mice. The use of females is absolutely relevant to dissect any possible gender effect of the diet in genomic instability.”**

Response: We appreciate the insightful comment from the reviewer. It is indeed crucial to consider gender-specific effects in studies involving diet-induced genomic instability. The choice of using male mice in our study was influenced by certain characteristics of the transgenic mutation reporter mouse model on an FVB background. Notably, FVB mice tend to exhibit lower

epididymal fat accumulation than C57BL/6 mice when fed high-fat diets, as reported in previous studies (Nascimento-Sales M, et al. *Physiol Rep.* 2017). In our preliminary diet studies, where both male and female mice were subjected to 10 kcal% control diet and 60 kcal% high-fat diets, we observed a distinct body weight separation over time in male mice compared to females. As a result of these observations, we opted to focus on male mice in this study. However, we are now backcrossing our mutation-reporter mice with C57BL/6 mice so that we can perform future studies in both male and female mice for gender comparisons.

We have now modified the Results section as follows:

High-fat diet induces obesity and liver steatosis.

The choice of using male mice in our study was influenced by the FVB genetic background of the transgenic mutation-reporter mice. FVB mice tend to exhibit lower epididymal fat accumulation than C57BL/6 mice when fed on a HFD⁵⁴. In our preliminary diet studies, we observed a more distinct body weight separation over time in males compared to females (data not shown) fed with the HFD, and therefore used only male mice in this study.

- 3) **“Are mutation frequencies in genomic DNA affected by short-term exposure to obesity? Moreover, based on the hypothesis of the authors, the DNA damage should be reversible in the liver, which is a tissue that has a high cell turnover rate. The authors should perform an experiment of changing to a low-fat diet to confirm this assumption.”**

Response: We value the reviewer's insightful comment. The focus of our study was to determine the impact of obesity on H-DNA-induced mutagenesis. We used a standard protocol to make the mice obese to test the hypothesis posed in this study. Additionally, upon returning obese mice to a control diet, we anticipate a reduction in further DNA damage and perhaps a reversal in some mutations but unlikely all mutations unless all the cells carrying mutations undergo a negative selection and are eliminated from the population. Large deletions would likely kill cells but smaller mutations (small deletions, small insertions and point mutations) would likely be fixed in replicating cells. DNA damage repair mechanisms may also be altered upon returning to a control diet in B-DNA mice, and the comprehensive understanding of this interplay might be influenced by factors such as the age of the mouse by the end of such a long study (Vaidya A, *PLoS Genetics*, 2014).

However, we acknowledge the importance of exploring short-term calorie restriction which has been shown by Ke Z, *npj aging*, 2020 to enhance DSB repair in various tissues in mice. While this is beyond the scope of the present study, we plan to conduct future experiments utilizing MMEJ/NHEJ reporter mice (Ke Z, *npj aging*, 2020) engineered to harbor B-DNA- or H-DNA-forming sequences in their genome. These mice could be subjected to a high-fat diet and a subsequent switch to a control diet. This approach might allow us to comprehensively explore the dynamic relationship between obesity, age, genomic instability and increased cancer risk.

Based on the reviewer's comment we have revised the Discussion section as follows:

While exploring the impact of transitioning from a HFD to a low-fat diet would be intriguing, the comprehensive understanding of this interplay might be influenced by factors such as the age of the mouse at the conclusion of a prolonged study. Moreover, mutations in and around alternative

structure-forming regions might be refractory to efficient repair, contributing to mutation accumulation and genomic instability. As expected, our results showed reduced DSB end-joining repair efficiency in tissue extracts from obese mice compared to those from normal-weight mice. While beyond the scope of the present study, future experiments utilizing NHEJ or MMEJ reporter mice⁸⁵ engineered to harbor B-DNA- or H-DNA-forming sequences and subjected to dietary interventions could provide comprehensive insights into the dynamic relationship between obesity, age, genomic instability, and cancer risk.

4) “Figure 6. The Western blots have been unclearly cropped (from the original files). The comparable samples should be loaded in the same blot to allow protein quantification.”

Response: We appreciate the reviewer's observation regarding Fig. 6. Confirming the reviewer's concern, we loaded samples from B-DNA mice (CD and HFD) on the same gel, and similarly, samples from H-DNA mice (CD and HFD) were loaded on the same gel.

It's important to note that in the initial experiment we included another mouse group subjected to a 30% calorie-restricted diet. However, due to inconclusive results on calorie restriction across different experiments, we opted to crop this group from the respective gels. The raw data, including the cropped-out samples, have been uploaded to Fig Share for transparency.

For clarity, in the original gels for B-DNA and H-DNA mice, the numbering was as follows:

B-DNA mice samples on gel:

1-5: Control diet

6-10: Calorie restricted diet

11-15: High-fat diet

H-DNA mice samples on gel:

1-4: Control diet

5-9: Calorie restricted diet

10-14: High-fat diet

We have now included the following sentence in the Fig. 7 legend:

For comparison, samples from B-DNA mice on CD and HFD were loaded on one blot, and samples from H-DNA mice on CD and HFD were loaded on a separate blot.

5) “How can this study be integrated in different human cancer risks associated to obesity?”

Response: This is a great question. This “proof-of-concept” study unveils for the first time, the impact of obesity on the mutagenic potential of an endogenous mutation hotspot, and demonstrates tissue-specific differences, laying the foundation for further investigation. To integrate these findings into the broader context of human cancer risks associated with obesity, several avenues could be explored:

- a) Genomic studies in neurological disorders: Given the colocalization of specific DNA sequences (capable of forming alternative structures) with neurological conditions, such as myotonic dystrophy and Alzheimer's disease, examining genomic instability of obese

individuals using Next Generation Sequencing could provide valuable insights into the heightened cancer risk associated with obesity.

- b) Exploring translocation-related leukemia and lymphoma risk: Investigating the *c-Myc* oncogene region, which co-localizes with alternate DNA structure-forming sequences, in obese patients could shed light on the increased risk of developing such blood cancers.

In summary, this study provides crucial insights into obesity-induced genomic instability while prompting avenues for targeted investigations that could enhance our understanding of the specific cancer risks associated with obesity in humans.

We have modified the Discussion section to include these points:

This study findings can be integrated in different human cancer risks associated to obesity. For example, it is worth noting that several alternative DNA structure-forming sequences have been implicated in various neurological disorders^{17, 18 19, 75}. Although we did not specifically examine different brain regions, exploring the association between H-DNA-forming sequences and neurological disorders such as Alzheimer's disease (AD) and dementia using NGS is warranted, considering recent reports linking obesity with these conditions⁷⁶. Understanding the potential involvement of H-DNA in such neurological disorders could provide valuable insights into their pathogenesis and open new avenues for their early diagnosis and treatment. Additionally, investigating the *c-MYC* oncogene region, which co-localizes with alternate DNA structure-forming sequences in obese patients could shed light on the increased risk of developing translocation-related leukemia and lymphoma.

Minor points

- 1) **“Figure 1 represents the research strategy and should be described in the methods section and moved to the Supplementary information.”**

Response: We appreciate the reviewer's suggestion regarding Fig. 1, which outlines our research strategy. Upon consideration, we opted to keep Fig.1 within the main figures. This decision stems from the necessity to highlight key aspects of the study design, particularly for readers less acquainted with alternative DNA structures and the mutation reporter-based transgenic mouse model. Thus, we think that including Fig. 1 in the main section enhances accessibility for a broader readership. However, we are open to relocating this figure to the supplementary section if the reviewer or editor deems it necessary.

- 2) **“Figure 2, please include stars in the panels.”**

Response: We have incorporated stars denoting significance levels in the panels. Additionally, we provide a legend that clearly outlines the significance level along with the corresponding p-value cutoff. This enhancement aims to improve the clarity and interpretability of the figures.

Reviewer #3 (Remarks to the Author):

“The manuscript, “Obesity increases genomic instability at DNA repeat-mediated endogenous mutation hotspots” by Kompella et al presents findings on the differential susceptibility of integrated H-DNA vs B-DNA sequences to point and deletion mutagenesis in mice on a high-fat vs standard chow diet. The investigation also probes underlying mechanisms that drive the increased mutation frequencies and the influences of end-joining processes under various physiological conditions. Although the overall conclusions seem largely supported by the data presented, in the following, there are several aspects of the investigation that need additional work and clarification”.

- 1) “In the Results section, additional data characterizing the founder and offspring B-DNA and H-DNA mice needs to be added. Given the procedures through which these mice were created, it would be anticipated that there could be significant differences in both copy number of integrations and the sites of integration. Authors should supply a more robust characterization of these strains. Further, it would be anticipated that there would be multiple founder mice for each genotype – were data obtained from multiple founders, and if so how comparable are the data within a genotype versus comparisons made between genotypes?”**

Response: We appreciate this comment from the reviewer. The transgenic mutation-reporter mice utilized in our study were established based on the methodology outlined in previous work by the Vijg group (Boerrigter METI, *Nature* 1995). Our lab further characterized this model, as documented in Wang G et al. *J Natl Cancer Inst.* 2008. The aforementioned paper details the characterization of multiple founder mice each carrying distinct copies of B-DNA- or H-DNA-forming sequences. Table 1 of this manuscript provides data indicating that despite the variations in copy numbers, the mutation frequencies within (B-DNA or H-DNA) and between (B-DNA vs. H-DNA) the mutation frequencies and spectra remain remarkably consistent. In the current study, B-DNA and H-DNA mice were selected based on the highest integrated reporter copy numbers (120-150), which is essential for successful pull-down blue-white mutation screening. We agree that using multiple funders that have different reporter integration copy numbers and integration sites is an ideal situation but doing so is financially not feasible. Notably, all the B-DNA and H-DNA mice on CD/HFD are identical in their reporter copy #s and integration sites and are perfectly comparable to each other. We trust that the additional information now provided addresses the reviewer’s concerns raised regarding potential disparities in copy number and integration sites.

Based on the reviewer’s comments, we have modified the Introduction section as follows:

The characterization of the founder mice published previously by Wang et al. (2008) showed that despite the variations in copy number, the mutation frequencies within (B-DNA or H-DNA) and between (B-DNA vs. H-DNA) the mouse type remain remarkably consistent⁵¹. In the current study, B-DNA and H-DNA mice were selected based on the highest integrated reporter copy numbers (120-150), which is essential for successful pull-down blue-white mutation screening. Thus, this

model represents a powerful tool for detecting DIO-related mutations across multiple tissues simultaneously⁵².

We have now included the following sentence in the Methods section on “Animal model and diets”:
The characterization of these mice has been previously published⁵¹.

- 2) **“Concerning the sequencing data for point mutations and deletions, of the 446 bp paired-end Illumina sequencing data that covered the B- and H-DNA regions, it would be useful to know the overall frequency within the H-DNA region.**

Also, the data in Fig 3 E (which appears to show point mutations even though the title to the figure legend indicates that this is for deletions) and S2 report bp positions 1-446, but the Tables 1-3 report frequency positions ranging from ~600 to ~1400 – it was not clear how to relate the data in the figures with the data in the tables.”

Response: We thank the reviewer for the effort they put into reviewing this figure and allowing us to expand further on the NGS data. We have calculated the overall mutation frequency in the 446-bp region and within the H-DNA region (253-276 bp of the amplicon) for mice on the control and high-fat diets presented in Fig. 3e and Supplementary Fig. 2. We have also provided details on the position of the amplicon with reference to the p2RT mutation reporter containing the H-DNA-forming sequence.

Based on the reviewer’s comments, we have modified the Results section as follows:

Obesity exacerbates mutations in H-DNA: insights from NGS analysis.

In addition to the *lacZ*-based mutation assays, we utilized Illumina MiSeq Next Generation Sequencing (NGS) on genomic DNA from mouse liver tissue as an orthogonal assay to assess mutations. This deep sequencing technology allowed us to quantify mutations within a 446-bp amplicon spanning the 686-1132 bp region of the p2RT mutation reporter, encompassing the H-DNA-forming sequence. Alignment of approximately 15 million NGS reads with the reference p2RT mutation-reporter sequence revealed a marked increase in percent non-reference alleles (comprising point mutations, insertions, and deletions) in and around the H-DNA-forming region. The increase was notably pronounced in the H-DNA mouse subjected to the HFD compared to the mouse on the CD, as shown in Fig. 3e. The overall percent non-reference alleles across the amplicon were 0.0094 in the mouse on the HFD contrasting with 0.0039 in the mouse on the CD. Within the H-DNA-forming region (253-276 bp of the amplicon) the mean percent non-reference alleles were markedly increased in the mouse on HFD the (0.0197) compared to the mouse on the CD (0.0060). These findings were corroborated by replicating the analysis in two additional H-DNA mice from each diet group (Supplementary Fig. 2). Similarly, across the amplicon, the mean percent non-reference alleles were 0.0076 in mice on the HFD versus 0.0048 in mice on the CD. In the H-DNA-forming region, this value was markedly elevated in the mice on the HFD (0.0197) compared to the mice on the CD (0.0057). These NGS-derived results also validate our mutation-reporter findings, indicating that obesity increases the mutagenic potential of this endogenous mutation hotspot *in vivo*.

We have modified the legend and text for Fig. 3e as follows:

Mutations mapped to the H-DNA-forming region.

Illumina Mi-Seq NGS deep sequencing across a 446-bp mutation reporter amplicon derived from genomic DNA from liver tissue of an H-DNA mouse on the CD and the HFD shows percent non-reference alleles. The blue bar represents the H-DNA-forming sequence region (253-276 bp) within the amplicon. The 446-bp amplicon corresponds to the 686-1132 bp region of the p2RT mutation reporter containing the H-DNA-forming sequence.

We have modified the legend and text for Supplementary Fig. 2 as follows:

Mutations mapped to the H-DNA-forming region.

Illumina Mi-Seq NGS deep sequencing across a 446-bp mutation reporter amplicon derived from genomic DNA from liver tissue of two additional H-DNA mice on the CD and the HFD shows percent non-reference alleles. The blue bar represents the H-DNA-forming sequence region (253-276 bp) within the amplicon. The 446-bp amplicon corresponds to the 686-1132 bp region of the p2RT mutation reporter containing the H-DNA-forming sequence.

- 3) **“For the section, “Obesity increases oxidative DNA damage in mice”, the text flows from Panels A & B to Panels I & J – the text description flows well, but it seems that the figure should be reorganized to flow Panels A-D. Data presented in the current Fig 4 Panels C-H should be a separate figure.”**

Response: We thank the reviewer for bringing this to our attention and apologize for any confusion. As described in our responses to Reviewer 1 (comments 6 and 11e) and this reviewer’s suggestion, we have reorganized Figure 4 for improved clarity. Fig. 4 now comprises panels a (8-oxo-dG: immunoassay), b (8-oxo-dG: Immunohistochemistry), c (OGG1: Western blotting), and d (OGG1: Western blotting quantification) to create a more cohesive flow. Additionally, previously Fig. 4 panels c-h, i (WB- γ -H2AX), and j (quantification of γ -H2AX) have been separated into a distinct Fig. 5. The original Fig. 5 has been renumbered as Fig. 6. The original Fig. 6 has been renumbered as Fig. 7. We believe this adjustment enhances the logical presentation of the data and improves overall figure readability.

- 4) **“The interpretation of data presented in Fig 5 D &E could be significantly influenced by data concerning mouse #5 (B-DNA mice on HFD) – the majority of the input substrate appears to have remained intact. It is also challenging that the signal/noise ratio in this assay is sufficient to report an 8.4% decrease in specific ligation products. It was also surprising that the overall efficiency of the ligation using DNAs with compatible ends did not appear to be significantly higher than reactions using DNAs with noncompatible ends (comparison of images in Fig 5 Panels D & G). The authors should provide insights into why these extracts do not efficiently catalyze compatible-end ligations. Overall, conclusions drawn from these data (“suggest that in the obese state, DNA end-processing of such intermediates may be compromised”) may be premature.”**

Response: We appreciate the reviewer's keen observation and the opportunity to clarify these points. Regarding the data concerning mouse #5 in Fig. 5 d & e, we acknowledge that the majority of the input substrate appeared to remain intact due to our inadvertent omission of the DNA

substrate incubation in the tissue extract. Consequently, we have removed mouse #5 from the figure for clarity and excluded it from our analysis.

Regarding the efficiency of ligation in compatible ends, we noticed that the kinetics of compatible end-joining was remarkably faster within the initial hour of substrate incubation in the mouse tissue extract. However, prolonged incubation (>4h) to promote the formation of longer products resulted in DNA loss. Conversely, with non-compatible ends requiring processing, the kinetics appeared slower, with maximum multimerization occurring by 4h. Further incubation in this case also led to loss of DNA. For consistency, we incubated the compatible and incompatible end substrates for the same duration of time (4h), ensuring fair comparison between the two conditions. During assay development, we consistently observed a close relationship between DNA loss and the rapid kinetics of end joining suggesting the initiation of some form of slow DNA 'degradative activity' in the extract upon exhaustion of the end-joining activity. Consistent with our results, similar findings were reported by Sathees et al *Mut Res* 1999.

We have added the following sentences in the Results section:

Obesity reduces the end-joining repair efficiency of DSBs in mice.

It is worth noting that the cell-free extracts from tissues contain active nucleases and other DNA repair/processing proteins, which may contribute to the degradation of both un-ligated substrates and ligated products. Although unavoidable, this is an indicator of the activity and functionality of cell-free tissue extracts. Therefore, we utilized testicular tissue extract for this assay, known for its high DSB end-joining efficiency and moderate nuclease activity compared to other tissues^{62, 63}. During assay development, we observed that the kinetics of compatible end-joining was remarkably faster within the initial hour of substrate incubation in the tissue extract. However, prolonged incubation (>4h) to promote the formation of longer products resulted in DNA loss. Conversely, for substrates with non-compatible ends requiring processing, the kinetics appeared slower, with maximum multimerization occurring by 4h and further incubation resulting in DNA loss. To ensure consistency, both compatible and incompatible end substrates were incubated for the same duration of time (4h), facilitating a fair comparison between the two conditions.

We have now modified the Discussion section as follows:

While beyond the scope of the present study, future experiments utilizing NHEJ or MMEJ reporter mice⁸⁵ engineered to harbor B-DNA- or H-DNA-forming sequences and subjected to dietary interventions could provide comprehensive insights into the dynamic relationship between obesity, age, genomic instability, and cancer risk.

- 5) **“Concerning the data presented in Fig 6, it seems unlikely that the integration of one or more linear plasmids (B- vs H-DNA mice) into the entire mouse genome would make any difference in the overall expression levels of proteins involved in DSB end processing/joining proteins – however, the differences that might be expected to be created under conditions of CD vs HFD would seem reasonable to test. Thus, even though the authors report some individual protein expression differences between B- vs H-DNA mice, limiting the analyses to such a presentation of the low vs high fat diet data seems more hypothesis driven and justifiable.”**

Response: We agree with the reviewer's comment and did not expect to see differences in protein levels in the H-DNA vs. B-DNA mice. Instead, we expected to detect differences in protein levels as a function of obesity in the mice. However, as we have previously demonstrated (Wang et al., *JNCI*, 2008), the H-DNA reporter integrated into the mouse genome, with a copy number of 10-20, exhibits inherent mutagenicity compared to the B-DNA reporter of a similar copy number. It is noteworthy, that in the current study, B-DNA and H-DNA mice were selected based on the highest integrated reporter copy numbers (120-150), so the mutagenic effects might be more significant than just one or a few DSBs. Moreover, our lab has shown that DSBs induced by H-DNA are repaired through a microhomology-mediated end-joining pathway, resulting in large-scale deletions with microhomologies at the breakpoint junctions (Wang & Vasquez, *PNAS*, 2004). Building on these findings and considering the increased mutation load in the obese H-DNA mice, we attempted to explore protein differences between the B-DNA and H-DNA mice under HFD conditions and found the results plausible. However, in future experiments, we plan to do systematic immunodepletion of DNA repair proteins in the testes tissue extract to gain deeper insights into the repair processes associated with H-DNA mediated genomic instability in the context of obesity.

Based on the reviewer's comments, we have modified the Introduction section as follows:

The characterization of the founder mice published previously by Wang et al. (2008) showed that despite the variations in copy number, the mutation frequencies within (B-DNA or H-DNA) and between (B-DNA vs. H-DNA) the mouse type remains remarkably consistent⁵¹. In the current study, B-DNA and H-DNA mice were selected based on the highest integrated reporter copy numbers (120-150), which is essential for successful pull-down blue-white mutation screening. Thus, this model represents a powerful tool for detecting DIO-related mutations across multiple tissues simultaneously⁵².

Minor

- 6) "The flow of information within the Introduction seems a bit disjointed – the text given between lines 90-114 seem to be at a basic general background level, and could be placed immediately after the opening paragraph, after line 44."**

Response: We appreciate the constructive feedback from the reviewer. Acknowledging the observation regarding the introductory flow, we have revised the text between lines 90-114. This content, which provides basic general background information, has been repositioned for better cohesion and now appears after paragraph 1 in the original manuscript. This adjustment aims to enhance the overall coherence and logical progression of the Introduction based on the reviewer's comments.

- 7) "In discussion of the characterization of mouse H-DNA sequences, the paper by Maekawa, K et al PNAS 119, 19 on mapping H-DNA sequences in the mouse genome needs to be added."**

Response: We appreciate the reviewer's suggestion and have added the reference: Maekawa K, Yamada S, Sharma R, Chaudhuri J, Keeney S. Triple-helix potential of the mouse genome. *Proc Natl Acad Sci U S A* 119, e2203967119 (2022).

We added the following sentences in the Introduction section:

In a recent study, the triplex-forming potential of H-DNA sequences in the mouse genome was mapped via S1-sequencing (S1-seq) using single-strand specific nucleases. Clusters of DSB-independent S1-seq signals were found to colocalize with H-DNA-forming sequences and strongly correlated with their potential to form a triplex motif ⁴⁹.

REVIEWER COMMENTS

Reviewer #1 (Remarks to the Author):

This submission is substantially improved by the new analysis and restructuring presented. I am grateful for the clarification of many points raised. I remain a little sceptical about the over-reliance on immunoblot quantification, and would hope the authors reflect on that for their extensions of this work. One point I think should be addressed further relates to my original point 4:

'There is an overall sense that the conclusion of this section is that H-DNA is more susceptible to 8-OxoG formation, and this might underlie the fragility of these sequences. This is untested in the manuscript and is not a not a conclusion that can be substantiated. It could be tested, at least in vitro, relatively easily'

The authors say they have tested this and will publish elsewhere. That is fine, but I think it is only fair to show the reviewers see this data and full experimental details, including how they validated the sequences are in the B and H conformations during the assay, to provide confidence that the assays performed satisfactorily address the point raised.

Reviewer #2 (Remarks to the Author):

The authors addressed my comments. Congratulations on the study.

Reviewer #3 (Remarks to the Author):

The revised manuscript by Kompella et al has comprehensively addressed all reviewers' comments and recommendations, resulting in a much clearer and impactful investigation. No additional changes or revisions are necessary prior to publication.

R. Stephen Lloyd

Response to Referees

Reviewer #1 (Remarks to the Author):

This submission is substantially improved by the new analysis and restructuring presented. I am grateful for the clarification of many points raised. I remain a little sceptical about the over-reliance on immunoblot quantification and would hope the authors reflect on that for their extensions of this work. One point I think should be addressed further relates to my original point 4:

'There is an overall sense that the conclusion of this section is that H-DNA is more susceptible to 8-OxoG formation, and this might underlie the fragility of these sequences. This is untested in the manuscript and is not a not a conclusion that can be substantiated. It could be tested, at least in vitro, relatively easily'.

The authors say they have tested this and will publish elsewhere. That is fine, but I think it is only fair to show the reviewers see this data and full experimental details, including how they validated the sequences are in the B and H conformations during the assay, to provide confidence that the assays performed satisfactorily address the point raised.

Response: We value the insightful feedback provided by the reviewer. We apologize for our inability to include the data in this manuscript, as they are included in another manuscript that is currently under review. Therefore, we think it is best to remove the paragraph asserting the claim that “H-DNA is more susceptible to 8-OxoG formation” from the manuscript. We trust that this revision has effectively addressed all outstanding issues.

For clarity, the discussion section was modified as follows:

Clinical studies have shown urinary 8-oxo-dG levels as a useful predictor of oxidative stress-mediated genomic instability during obesity^{77, 78}. We observed a significant increase in 8-oxo-dG levels in obese mice compared to normal weight mice. Previous studies by the Llyod (2012) and Sampath (2018) groups have demonstrated the susceptibility of the 8-oxo-dG repair enzyme, OGG1-deficient mice to adiposity and steatosis under HFD conditions⁷⁹, while overexpression of mitochondrial OGG1 offers protection against diet-induced obesity⁸⁰. In our study, we observed an upregulation of OGG1, likely a protective, albeit insufficient, response aimed at mitigating obesity-induced oxidative DNA damage. Remarkably, these results are consistent with a recent study reporting upregulation of the OGG1 gene in the visceral adipose tissue of individuals with obesity⁸¹.

Oxidized DNA can result in SSBs via direct⁸² and indirect mechanisms⁸³. Indeed, the increase in SSBs was significant in obese H-DNA mice compared to the mice with B-DNA-forming-sequence in their chromosomes. Additionally, we have previously demonstrated that H-DNA can stimulate the formation of DSBs via error-generating processing in eukaryotic genomes⁵³. Our study provides evidence of obesity-induced stimulation of DSBs at an endogenous mutation hotspot, and of DNA damage responses, as indicated by increased levels of the DSB

marker γ H2AX in testicular tissue extracts. These observations are consistent with a recent study showing elevated γ H2AX foci in small follicles of ovaries from obese vs. lean mice⁸⁴. We have previously reported the occurrence of DSBs induced by H-DNA structures in normal-weight mice⁵⁰. However, under obese conditions, the DSB levels and, subsequently, γ H2AX levels were found to be significantly amplified.

Reviewer #2 (Remarks to the Author):

The authors addressed my comments. Congratulations on the study.

Response: We thank the reviewer for their time and consideration.

Reviewer #3 (Remarks to the Author):

The revised manuscript by Kompella et al has comprehensively addressed all reviewers' comments and recommendations, resulting in a much clearer and impactful investigation. No additional changes or revisions are necessary prior to publication.

R. Stephen Lloyd

Response: We sincerely thank the reviewer for their time and valuable feedback.